# An Optimal Metro Design for Transit Networks in Existing Square Cities Based on Non-Demand Criterion

**Mahmoud Owais [1] , Abdou S. Ahmed [1],*, Ghada S. Moussa [1] and Ahmed A. Khalil [2]**

1   Civil Engineering Department, Assiut University, Assiut 71515, Egypt; maowais@aun.edu.eg (M.O.);
    ghada.moussa@aun.edu.eg (G.S.M.)
2   Civil Engineering Department, Faculty of Engineering at Shoubra, Benha University, Shoubra 11629, Egypt;
    ahmed.khalil@feng.bu.edu.eg
*   Correspondence: Abdouahmed@aun.edu.eg; Tel.: +20-1066002203

**Abstract:** The overall purpose of this study is to enhance existing transit systems by planning a new underground metro network. The design of a new metro network in the existing cities is a complex problem. Therefore, in this research, the study idea arises from the prerequisites to get out of conventional metro network design to develop a future scheme for forecasting an optimal metro network for these existing cities. Two models are proposed to design metro transit networks based on an optimal cost–benefit ratio. Model 1 presents a grid metro network, and Model 2 presents the ring-radial metro network. The proposed methodology introduces a non-demand criterion for transit system design. The new network design aims to increase the overall transit system connectivity by minimizing passenger transfers through the transit network between origin and destination. An existing square city is presented as a case study for both models. It includes twenty-five traffic analysis zones, and thirty-six new metro stations are selected at the existing street intersection. TransCAD software is used as a base for stations and the metro network lines to coordinate all these data. A passenger transfer counting algorithm is then proposed to determine the number of needed transfers between stations from each origin to each destination. Thus, a passenger Origin/Destination transfer matrix is created via the NetBeans program to help in determining the number of transfers required to complete the trips on both proposed networks. Results show that Model 2 achieves the maximum cost–benefit ratio (CBR) of the transit network that increases 41% more than CBR of Model 1. Therefore, it is found that the ring radial network is a more optimal network to existing square cities than the grid network according to overall network connectivity.

**Keywords:** grid; ring-radial; planning; metro network; passengers transfer; transit systems

---

## 1. Introduction

Metro transit systems are a vital strategy to mitigate the traffic congestion problem in existing big cities. Thus, this paper aims to design a new metro network to increase the overall transit system connectivity by minimizing passenger transfers. In this article, two models of a transit system are built in an existing urban city with a grid or ring-radial transit network to find the optimal network. The optimal network is found to increase direct trips and minimize passenger transfer between origin and destination.

To achieve comprehensive results for servicing existing cities of different sizes, this study idealizes existing square cities with non-demand criterion. The research methodology also shows how to use this criterion to achieve the optimal network planning of transit systems for these cities. The research methodology achieves the overall network connectivity with minimum construction cost (the best

cost–benefit ratio). Moreover, theoretical concepts are selectively used, particularly from graph theory, focusing on those with practical relevance to actual metro network planning.

We aim to develop a future scheme considering the non-demand criterion in metro network design in existing cities. However, it is a complex task when metro stations are fixed locations at an existing street intersection. More specifically, there are four major challenges, which need to be overcome in this paper.

(1)  How to design a ring-radial network in existing square cities
(2)  How to identify the passenger transfer between stations
(3)  How to measure passenger transfer's effects on the metro network design of both models
(4)  How to minimize passenger transfers and increase the cost–benefit ratio

There are three points to address in the optimization of the passenger transfer number (PTN): (a) defining an objective function that accurately represents the connectivity of the overall transit network, (b) developing an algorithm to minimize PTN among public transportation facilities, and (c) creating a passenger transfer matrix via the NetBeans program to help in determining the PTN required to complete the trips on the metro transit network.

The structure of the paper is as follows: Section 2 presents the literature review. Section 3 provides input data, models, and assumptions. Section 4 defines the hypothetical grid city as a case study. In Section 5, the analytical model of design is illustrated with the results and discussion. Section 6 presents the conclusions.

## 2. Literature Review

Today, millions of passengers rely on the metro transportation system in their daily commuting. Big existing cities with rapid population growth and a continuous acceleration of the urbanization process often face more severe traffic problems than other smaller cities [1,2]. In some of these cities, the transportation system cannot satisfy users' current and future needs [3–5].

Therefore, the optimal planning of public transportation networks leads to a substantial improvement in the efficiency of the urban transportation network [6], whereas the transit network planning incorporates complex decisions at the strategic, tactical, and operational stages of design [7]. The problem becomes more complicated when the transit network was established long ago with a physical structure that is difficult to change. Consequently, the planning problem would be transformed into an improving problem. Improving transit networks includes building new lines with a well-connected structure that have a self-adaptive ability to recover from transport network disruptions.

For transit systems to be attractive to car users, they have to fulfill a minimal level of service. Improvement in network topology and operational management is the main element to redirect the transit situation and enable more efficient and effective systems. One study presented the radial hybrid model to extend its applicability for the other major group of regular cities, radio centric cities where these cities' street pattern is composed of radial axes and concentric rings [8].

More recently, ring-radial networks naturally concentrate passenger's shortest paths, and to the economies of demand, the concentration that transit exhibits. Thus, it appears that ring-radial street networks are better for transit than grids [9].

Regarding the road layout, existing urban cities are classified according to their street pattern, such as (i) irregular cities whose street network does not present any kind of order and (ii) regular cities, which are defined by a characteristic pattern of streets. There are mainly two patterns in regular cities, grid and radial structures [10,11]. In this study, regular square cities with existing grid streets were proposed for both models.

Transit systems must uniformly cover the service region in space and time with well-spaced transit stops and frequent reliable service. Good spatial coverage limits the walking time to/from every point in the service region, and good time coverage limits the waiting and transfer times [12,13]. One study examined the structure of urban transit systems that can deliver an accessibility level

comparable to that of the automobile and the cities' character in which this can be done at a reasonable cost. These transit networks should provide good service between every pair of points in the city throughout the day and be easily understood by the public. They found that increasing the spatial concentration of stops beyond a critical level increases both the user and agency costs. In many cities, transit networks have been built sequentially and do not fit the users' needs anymore. The results are long travel times and an unnecessarily high number of people who have to transfer [14,15].

The planning of metro networks is usually a complex problem. Consideration of local conditions, such as demand characteristics, the existence of transportation corridors, requirements for specific station locations, and so on, tend to suppress the analyses of network topology and geometric characteristics. However, even though designers of metro lines and networks may want to use some design guidelines, perform comparative analyses, or use experiences from network operations in other cities, because of the high cost of metro construction and the permanence of its facilities, it is essential to design optimal networks with respect to service for passengers, the efficiency of operation, and the metro system's relationship to the city. This is a complex task, and it deserves more attention than it has received until now [16]. An attempt is made in this study to provide design guidelines that may assist in the planning and design of metro system networks, lines, and stations.

Rail transit networks can be classified by their geometric forms into several types. Transit network forms can be classified into several general types, depending on street networks, urban form (land-use patterns, densities, etc.), topography, and several other factors. Each type has some characteristic features [17–20]. In previous research, grid networks are defined as networks consisting of parallel and rectangular lines, usually following a grid street pattern. These networks provide better area coverage and less focus on a single point than radial networks. Besides, radial networks consist of radial and diametrical lines meeting or intersecting in the city center. They are sometimes supplemented by a circumferential or ring line. These networks concentrate on the city center and tend to have high peaks because of the large number of commuters they carry [21].

The objective of the "non-demand" criterion for transit network design is to maximize the direct trips between OD pairs, which increases the overall network connectivity. This criterion of the design is based on both the topological layout of the transit network and the demand center's distribution and does not depend on the uncertain information about the number of trips generated between each demand pair. It would reflect on the intensity of passenger flows, their direction, distribution by travel goals, and distribution over daytime, which, as mentioned, is uncertain information in nature. In metro cases, practitioners could easily treat capacity (unlike bus networks) in the operational stage by setting the reasonable train vehicle capacity and number.

Passenger transfers between transit routes or modes represent an essential component of transit travel. No transit network can serve all trips by direct routes without any transferring. Actually, the more transferring is performed, the easier it is to operate different routes efficiently, each one specifically designed to its physical conditions, volume, and character of demand. Transfers do, however, cause an inevitable delay in a passenger's travel; they also require some walking, orientation, and other actions that may involve time and effort [18,19]. Connectivity is a significant problem in large-scale transit networks, because the number of transfers required to conduct a trip is considered a discomfort by transit users. In [22], the authors presented a practical solution for an underground metro line planning problem by integrating existing bus and metro networks into a single connected transit network. According to predefined demand node pairs, one single metro line is designed to minimize passenger transfers through the transit network. After that, the design scheme has offered a set of ring route alternatives for the sizeable case study in Greater Cairo.

The planning, facility design, and scheduling of transfers are of great importance for both transit system efficiency and user convenience and attraction. If a transit system provides easy, simple, fast, and convenient transfers, its entire network can be operated very efficiently, and it can attract most potential users. If, on the other hand, transfer locations are poorly designed, unsafe, and unpleasant, and schedules are not coordinated, transferring may be such a serious obstacle, that deters many

potential passengers from using transit services. Metro is one of the largest public investments that a city makes and has substantial permanent impacts. Therefore, a thorough understanding of various geometric forms of the metro network, and their inherent functions and operational characteristics must be an important factor in metro network planning network forms that perform in creating an efficient metro network system in any city [17,20,23,24].

Planning is a function not only of analyses and strategies, but also of visions and ideas. It is complex and intricately involved at many levels and often involves qualitative information that is not always easy to define or analyze. Furthermore, many aspects of metro planning vary greatly depending on specific local conditions and are sensitive to inherent local political climates.

With the expansion of the metro network in big cities, transfer efficiency at transfer stations is of vital importance. To improve transfer efficiency, operators need to coordinate the arrival and departure time of trains at transfer stations to implement smoothly transferring as much as possible [25]. Transfers have several attributes that make them incredibly inconvenient, such as the discomfort of boarding another mode, and the negative perception of waiting for the arrival of that transfer vehicle [26]. One study defined transfer optimization as minimizing the overall inconvenience to passengers who must transfer between lines in a transit network.

In a metropolitan area, the transit network cannot economically provide direct services among all origins and destinations, especially when and where demand densities are low. Thus, a transit network may rely on routes connecting at transfer stations, and passengers may have to transfer and wait for the connecting vehicles at those transfer stations [27]. In public transportation networks, most operations will require some transfer movements from one route to another to serve the diverse origin–destination patterns. Therefore, transfers of passengers among routes are used to (1) obviate the need for direct routes connecting all origin-destination pairs and (2) concentrate passengers on major routes with high speed (and high cost) equipment. To minimize the adverse effects, the transit operator must try to (1) reduce the required number of transfers and (2) minimize the transfer delays (i.e., the waiting time at the transfer stations) [27].

A transit system is an integral part of every urban transportation system. Since the construction of transit systems in big cities requires great investment, the use of a currently operating transit system can help transportation authorities to better manage existing demands [28]. They study the transit network scheduling problem, which aims to minimize the waiting time at transfer stations. A public transportation network is one of the basic components of transit system planning. One study proposed a hybrid optimization model for the urban bus transit route network design problem (TRNDP). They concluded that the total travel time for the proposed method was significantly lower than that of the competing method, with a 21.51% reduction. In addition, the proposed method provided 85.23% direct travelers, 14.65% travelers with one transfer, 0.12% travelers with two transfers, and no unsatisfied demand [29].

Transit route network (TRN) design is an important design component in the transit planning process, which also involves a transit network schedule (TNS) design and consideration of issues related to transit policies. TRN design is mainly concerned with the layout of transit routes and the determination of transit stops. The quality of a TRN may be evaluated in terms of several network parameters, some of which are route directness, service coverage, network efficiency, and several transfers required [30]. They present a mathematical methodology for transit route network optimization by providing an effective computational tool for the optimization of a large-scale transit route network to minimize transfers and maximize service coverage.

Metro is a common kind of rail transit system, where the metro transfer station is an important node of the rail transit network. Thus, the metro transfer station in china is always a limited space with crowded passengers. Currently, the intelligent transport system and advanced public transportation system have been applied to the metro operation and management, but conflicts of passengers remain a problem, and passenger waiting time is still very long [31]. They deal with the metro passenger transfer problem through the coordination of different metro lines. They established metro transfer

optimization models including three scenarios: (1) single transfer station converged by two metro lines with the same scheduling plan; (2) single transfer station converged by two metro lines with different scheduling plans; (3) multiple transfer stations converged by two metro lines with different scheduling plans. The results show that this paper's models can provide a coordination scheme of different metro lines to alleviate passenger conflicts and reduce passenger waiting time in the transfer station.

Metro transfer station refers to a place where passengers from different lines gather together and passengers can transfer from one line to another without having to buy another ticket. Metro transfer station is an important node of the urban rail transit network [31]. The transfer is considered as the most inconvenient factor by the public passengers, which usually means that they have to walk and wait (usually queuing). In the existing public transit system, the unnecessary transfer should be reduced, which means reducing the average transfer times [32].

Transit network optimization is developed to minimize the total time cost of all the trips [33]. They studied the effect of the transfer time composition on the total time expense of a transit trip and proposed a genetic algorithm to solve the newly developed optimization model. Public transit planning comprises a variety of interesting optimization problems. In network planning, the public transit network optimization steps are lines or routes that connect stops on the network with certain frequencies [34]. They introduce a new optimization approach for designing a public transit network, and the main innovation is that the demand for public transit is endogenous. Transit network design is a very important problem. It has a significant influence on passenger satisfaction with the whole transit network system [35].

One of the inherent difficulties of designing a transit network is the need to offer a desirable level of service for passengers while ensuring a profitable operation for the transit agency. From the passengers' standpoint, the service should be fast, compared to the other transportation modes available to them, and should mainly accommodate trips directly, i.e., usually up to one or longer trips; a maximum of two transfers between vehicles can be tolerated [36]. Transportation engineering poses a multitude of optimization problems that are extremely difficult to solve using traditional mathematical programming techniques. One such problem is the urban transit network design problem (UTNDP), where routes and schedules for an urban transit network have to be developed efficiently [37].

A sustainable transportation system must include a robust public transportation system (or transit system); building more roads for the ever-increasing number of private vehicles (automobiles) is not the solution. However, for a public transportation system to be successful in attracting passengers, the system must be efficient [38]. The urban transit routing problem (UTRP) is a highly complex multi-constrained problem, and the evaluation of candidate route sets can be both time-consuming and challenging, with many potential solutions rejected on the grounds of infeasibility [39].

The line-planning problem is one of the fundamental problems in the strategic planning of public and rail transport. The transport company wishes to minimize its operating cost; the passengers request short travel times [40]. In urban metro systems, stochastic disturbances repeatedly occur due to an increment of demands or travel time variations; therefore, improving the service quality and robustness by minimizing the passengers waiting time is a real challenge [41]. They proposed a two-stage Genetic Algorithms GA-based simulation–optimization approach to minimize the expected passenger waiting times.

In several cities, metro systems provide a desirable alternative to private transportation. They reduce traffic congestion, improve mobility, and contribute to the protection of the environment. Designing a metro network is a complex problem in which several criteria must be considered, technical difficulties are important, and the costs are very high. These problems also involve many different decision-makers, such as city planners, engineers, interest groups, and politicians [42].

The process of planning the metro system is very complex. This is due to the large amount of required input data originating from many domains. Additionally, the required extraordinary budget makes the design phase difficult and forces some trade-offs to meet most expectations. Designing



a metro system often requires a multi-variant analysis to examine possible future ways of urban agglomeration development [43].

Most of the line design techniques depend on the demand information, which adds more complexity to the problem. As the conventional transportation problem, the transit O/D estimation could be based on the well-known step planning models (i.e., trip generation, trip distribution, modal choice, and traffic assignment) [44–47]. For the actual size networks as our case, the four models are unlikely to give accurate results due to the high level of uncertainty at the operational stage [48,49].

Once the transit network structure is defined, the strategic level is followed by two other consecutive stages. The tactical stage tackles frequency setting and capacities determination, whereas the operational stage includes timetables, train schedules, and crew scheduling [1,50,51]. The transit route/line network design problem's (TRNDP) complexity, besides its several aspects of design, leads to solving each stage separately [52,53]. Any attempt to solve them simultaneously requires many relaxation assumptions [54–57].

Considering the line design in the strategic stage, three solution approaches are found in the literature, namely, integer linear programming, meta-heuristics, and analytical models. In the first approach, the link selection for a line becomes a binary variable (0 or 1) in a mathematical formulation representing the objective function of design while subjecting to some planning constraints [40,58]. Usually, these models are applicable to a small- and medium-sized network and intractable for large scale networks. Meta-heuristics are used alternatively to incorporate larger networks, while they use neighborhood search operators to offer approximate solutions [59,60]. Many meta-heuristics algorithms are adapted to the problem like simulated annealing (SA) [61], Genetic Algorithms (GA) [37], tabu search (TS) [59], and swarm intelligence (SI) [62]. The analytical models obviate the introduction complexity of any new track using the available spatial, economic, technical, social, and political information. They depend on suggesting pre-structured routes such as radial lines, diametrical lines, branch lines, gridlines, and ring lines [9,63–67].

Many other cities in China and elsewhere also envision ring transit lines for their future rail transit networks. Beijing currently has two full ring lines surrounding the central business district (CBD), making the connections between lines more convenient and easier [64]. Additionally, Paris is currently constructing its second ring line on the city outskirts with a completion deadline in 2025 [68]. In the case of multiple ring lines alignment, it should be noticed that an existing ring line would affect the optimal alignment of the second ring line [64].

Ring lines are positioned around a city center and intersect radial lines to create transfer opportunities; therefore, they support an integrated multipath network. In addition to saving passenger travel and wait times, increasing transit network connectivity and reliability, and reducing the transit load in the downtown core, ring lines also have the potential to increase the accessibility and development of new satellite centers [64]. In [66], three important functions of ring lines are listed: (a) improving connectivity between radial lines and distributing the congestion away from radial lines in the CBD, (b) making trips between radial lines shorter by bypassing the CBD, and (c) serving the busy areas in a ring around the CBD.

To this end, the literature review presented different characteristics of the transit design problem. However, to the best of our knowledge, a few previous studies have an optimization model devoted to using a non-demand criterion. Therefore, it is a valuable study for researchers and planners so as to drawing preliminary conclusions to transport network planning. Finally, minimizing the numbers of passenger transfers between the OD pairs is capable of simultaneously reducing transfer waiting and improving transfer availability that leads to decreasing the traffic congestion in existing cities.

## 3. Input Data, Models, and Assumptions

In this study, we present an analytical model of metro network design using a non-demand criterion objective function of the passenger transfer number (PTN) for improving the coverage

performance of the existing city networks. The explanation of the mathematical formulation could be summarized in the following points;

- Introducing a mathematical formulation for determining the PTN of the metro transit network.
- Obviating the combinatorial complexity of transit routing with an efficient, straightforward scheme of design.
- Presenting the mathematical notations in a general framework, which gives the possibility of generalizing the methodology in other studies or even using it as a subroutine in presenting more sophisticated methodologies.
- Developing an exclusive non-demand criterion of metro network design, unlike reviewed studies where they all are demand-oriented.

This section describes the input data, models, and assumptions. Section 3.1 defines network representation. Then, Section 3.2 presents transfers in transit networks. Finally, Section 3.2 states objective function.

### 3.1. Network Representation

For an urban transport network, it is defined as non-directed graph $G = (V, E)$, where $V$ is the set of vertices (nodes) that are connected by the set of edges. The set of $E$ is none empty set = $\{(s, e): s, e \in V, c_{i\text{-}j} \neq \infty\}$, in which each node pair $(s, e)$ is a bi-directed arc (edge, link) with cost $c_{s\text{-}e}$ (travel time or aggregate impedance) [69]. The $W$ set contains every demand pair $(o/d)$. Each node $(s)$ has $ADJ(s) = \{e \in V: (s, e) \in E\}$ set of the nodes that are connected directly with one edge. The metro network is defined by a set of lines $L = \{l_1, l_2, l_i, \ldots l_j, \ldots l_k \ldots, l_n\}$. Each line consists of a set of connected edges without any internal loops $\{((o,.), \ldots, (s, e), \ldots, (.,d)): o/d \in W, o,s,e,d \in V\}$. Note that all nodes are considered the potential metro stops.

### 3.2. Transfers in Transit Networks

In transit networks, passengers do not always find a direct line from their origin to their destination. That is when they need to use more than one line. Typical types of transfers in a transit network are shown in Figure 1.

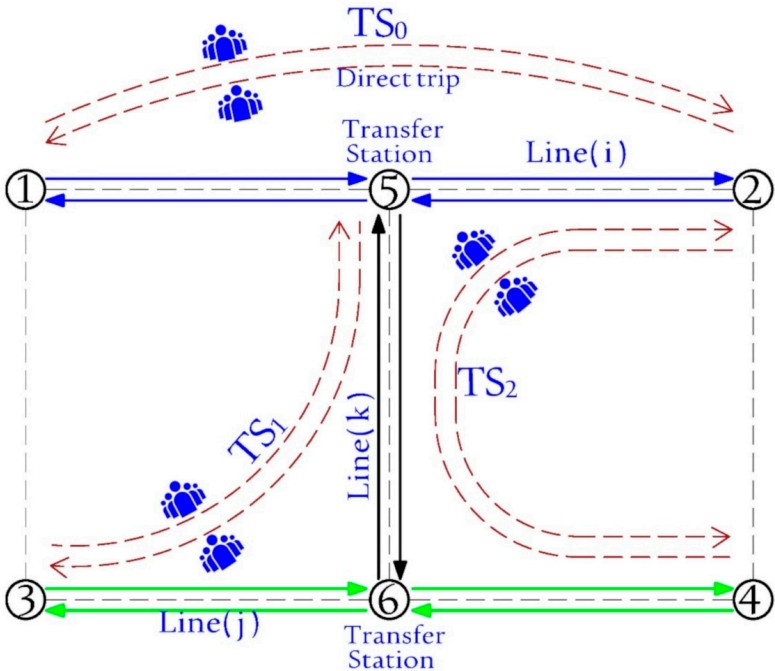

**Figure 1.** Definitions of passenger transfers between stations.

In Figure 1, ($TS_0$) means that the passengers travel from the origin to the destination using direct trip without transfer stations (TS = 0). ($TS_1$) means that the passengers move from the origin to the destination using two trips with one transfer station (TS = 1). ($TS_2$) means that the passengers travel from the origin to the destination using three trips with two transfer stations (TS = 2). Finally, if none of these cases are found, a not-served demand pair is declared, where passengers cannot commute from the origin to the destination with the existing metro network ($TS_\infty$).

### 3.3. Objective Function

A generalized objective function is developed to represent the connectivity of the overall transit network. At the line network level, the number of transfers is evaluated in terms of the number of demand node pairs covered directly or indirectly by the lines itinerary [69]. The proposed objective function tries to minimize the number of potential total number transfers for the existed system. It can be defined as follows:

Minimize:

$$PTN = \sum_{o,d \in W} (-\beta_0 TS_0 + \beta_1 TS_1 + \beta_2 TS_2 + \beta_3 TS_\infty) \tag{1}$$

$$s.t. \forall\, o, d \in W$$

$$TS_0 = \sum_{i \in L} \alpha_{io} \alpha_{id} \tag{2}$$

$$TS_1 = \left[ \sum_{i \in L} \sum_{j \in L} (\alpha_{io} \alpha_d) \left( \sum_{m \in V} \alpha_{im} \alpha_{jm} \right) \right] (1 - TS_0) \tag{3}$$

$$TS_2 = \left[ \sum_{i \in L} \sum_{j \in L} (\alpha_{io} \alpha_{jd}) \left( \sum_{m \in V} \alpha_{im} \alpha_{km} \right) \left( \sum_{u \in V} \alpha_{ku} \alpha_{ju} \right) \right] (1 - TS_1) \tag{4}$$

$$TS_\infty = \begin{cases} 1, & if\ TS_0 + TS_1 + TS_2 = 0 \\ 0, & otherwise \end{cases} \tag{5}$$

$$i,\, j,\, k\ \in L,\ and\ i \neq j \neq k,$$

where:

$TS_0$ = the numbers of passenger transfers between OD pairs without a transfer station.

$TS_1$ = the numbers of passenger transfers between OD pairs with one transfer station.

$TS_2$ = the numbers of passenger transfers between OD pairs with two transfer stations.

$TS_\infty$ = the numbers of passenger transfers between OD pairs with more than two transfer stations or unserved areas.

$\alpha_{ls}$ = incident symbols equal to 1 if metro line (*l*) passes the node (*s*), 0 otherwise.

$o, d$ = the origin, destination nodes,

$m$ and $u$ = transfer nodes.

$\beta$ = a non-negative weight factor reflecting the relative importance of each component in the minimization process.

The method of selecting the coefficients in the OD matrix format as following:

$\sum TS_0$ = the sum of matrix cells that are covered directly with the existing transit system (with zero transfers).

$\sum TS_1$ = the sum of the cells that are covered with one transfer.

$\sum TS_2$ = the sum of the cells that are covered with two transfers.

$\sum TS_\infty$ = the sum of all not served demand node pairs.

In Equation ((2, $TS_0$ would equal 1 or higher if nodes (*o*) and (*d*) are connected directly by one or two transit lines. If $TS_0$ is a non-zero value, it is set to 1, and the OD pairs are designated as covered with zero transfers. Otherwise, we calculate $TS_1$. In Equation (3), the first term checks that there are

simultaneously a transit line (*i*) passing by (*o*) and a transit line (*j*) passing by (*d*), whereas the second term checks that there is a shared transfer station (m) between the two lines.

The last term stipulates that if the OD pairs are classified before as non-transfer ($TS_0 = 1$), it cannot be classified as one transfer at the same time. Moving to calculate $TS_2$ is similar to $TS_1$ and its value is also set like $TS_1$. If all are zeros, these OD pairs are set to the non-covered category ($TS_\infty = 1$). In objective (1), the only term to maximize is $TS_0$, which matches the logic of the design in minimizing transfers and maximizing direct trips.

## 4. Metro Models Concept

This section presents the models and analysis methods. Subsection A introduces basic assumptions. Then, subsection B presents the metro network structure.

### 4.1. Case Study

This subsection presents the existing square city of the assumptions considering fixed stations for both models. An existing city is assumed to be square with Width "W" (equal to Length "L") and to have a square grid street with street spacing "S", as shown in Figure 2. It is assumed in this paper that the three main items are as following:

- Stations' locations are fixed positions in main street intersections, as similar in existing cities.
- Existing street spacing (S) is equal to a distance unit (1 km).
- Zone traffic area is assumed to be between existing streets and equal to an area unit (1 km$^2$)

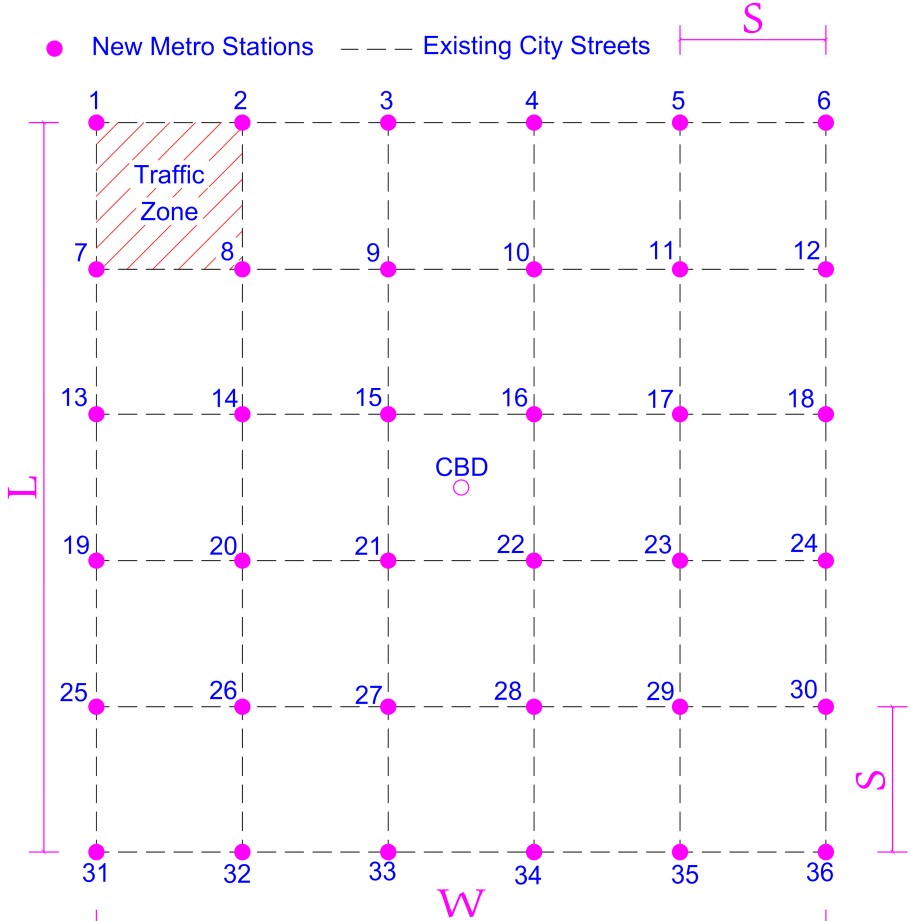

**Figure 2.** An existing square city with new proposed metro stations.

### 4.2. Metro Network Structure

This subsection introduces a representation of the assumptions concerning the structure of the metro transit networks for both models. It is assumed for Model 1 that the metro network has a square grid system. Moreover, a central business district (CBD) is assumed to be in the middle center of an existing city. In this model, the grid metro network is achieved by metro lines in one or both directions. As in Figure 3, metro lines are assumed to be in one direction (L1, L2, L3, ..., L6). Besides, metro lines are assumed to be in two directions (L1, L2, L3, ..., L12), as shown in Figure 4.

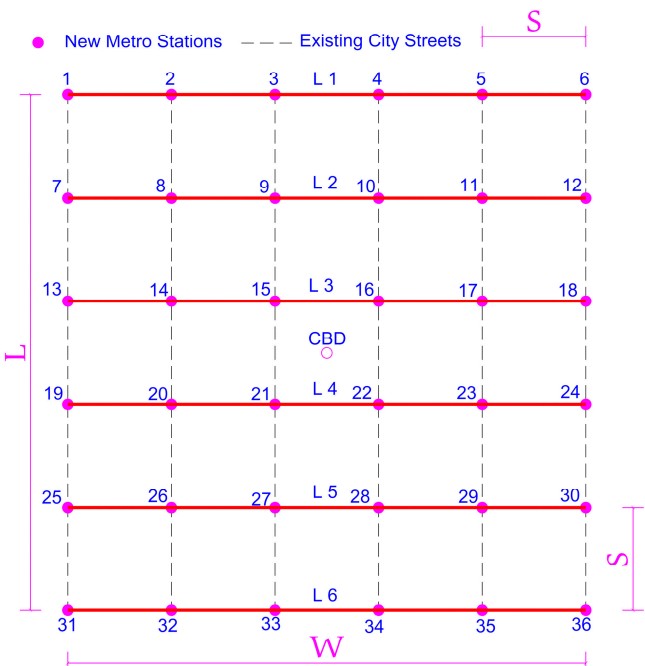

**Figure 3.** Grid metro lines in one direction (Case 1).

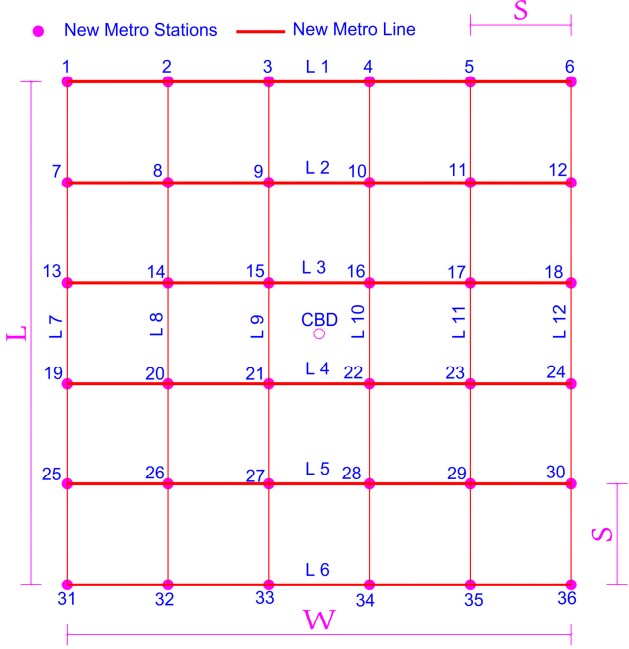

**Figure 4.** Grid metro lines in both directions (Case 2).

It is assumed for Model 2 that the metro network has a ring-radial system. Moreover, a central business district (CBD) is assumed to be in the middle center of an existing city. In this model, network planning is adapted to the existing urban structure with ring lines and radial lines connections, whose corridors are focused on the existing main streets

As in Figure 5, metro lines are assumed to be in ring direction (L1, L2, L3), and the radial direction (L4, L5). Additionally, in the same figure, metro stations are assumed to be fixed location in main street intersections. Additionally, modification of some metro paths is achieved, as shown in Figure 6.

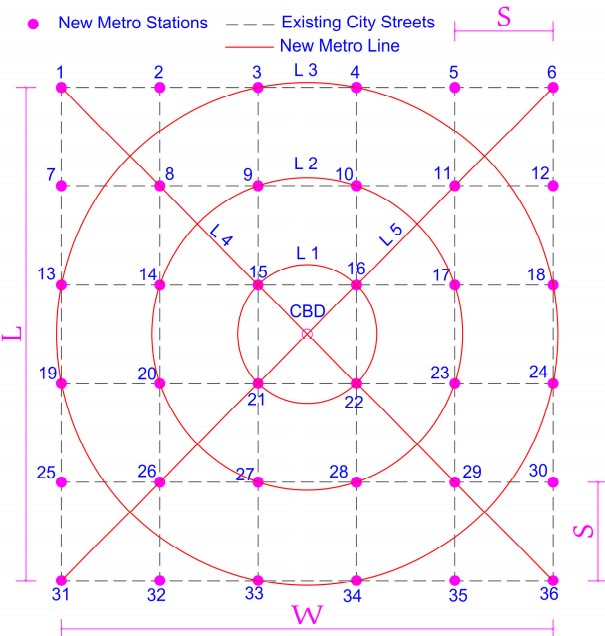

**Figure 5.** Ring-radial metro lines in the existing square city (Case 3).

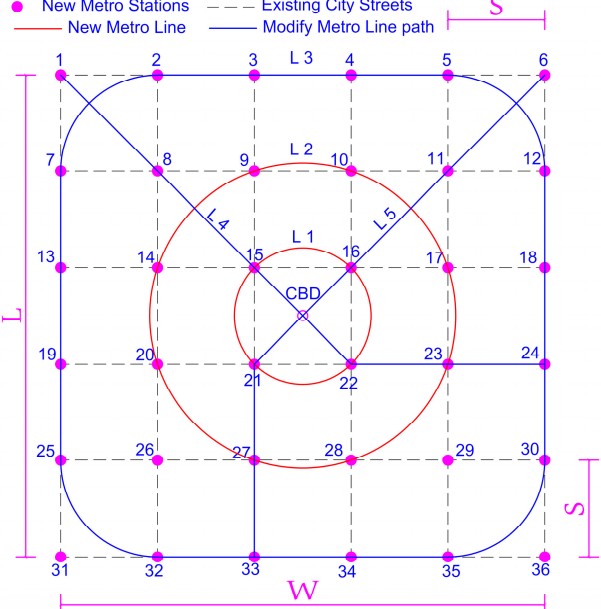

**Figure 6.** Ring-radial metro lines in the existing square city with modification of some metro paths (Case 4).

## 5. The Analytical Model of Design

The general configuration of the model consists of three phases, as follows:

Phase #1:    Make an assessment and analysis of the existing metro according to different PTN structures.

Phase #2:    Obtain the passenger transfer matrix of both models.

Phase #3:    Show the efficiency of the PTN criterion as a means of design to select the optimal metro network.

### 5.1. Assessment of the Transit Network

In the first phase, TransCAD is used to locate the existing street within the study area, as well as to identify the proposing metro lines and stations. This phase of the model is dedicated to the description of the proposed OD transfer matrix estimation methodology, which is based on the PTN between stations, whereas the passenger OD transfer matrix is created by the NetBeans program with the Java programming language [70]. An OD transfer matrix consists of 36 rows × 36 columns, and the summation value of the matrix cells for each category of a transfer needs to be estimated. The algorithm for that task consists of the following steps, as shown in Figure 7.

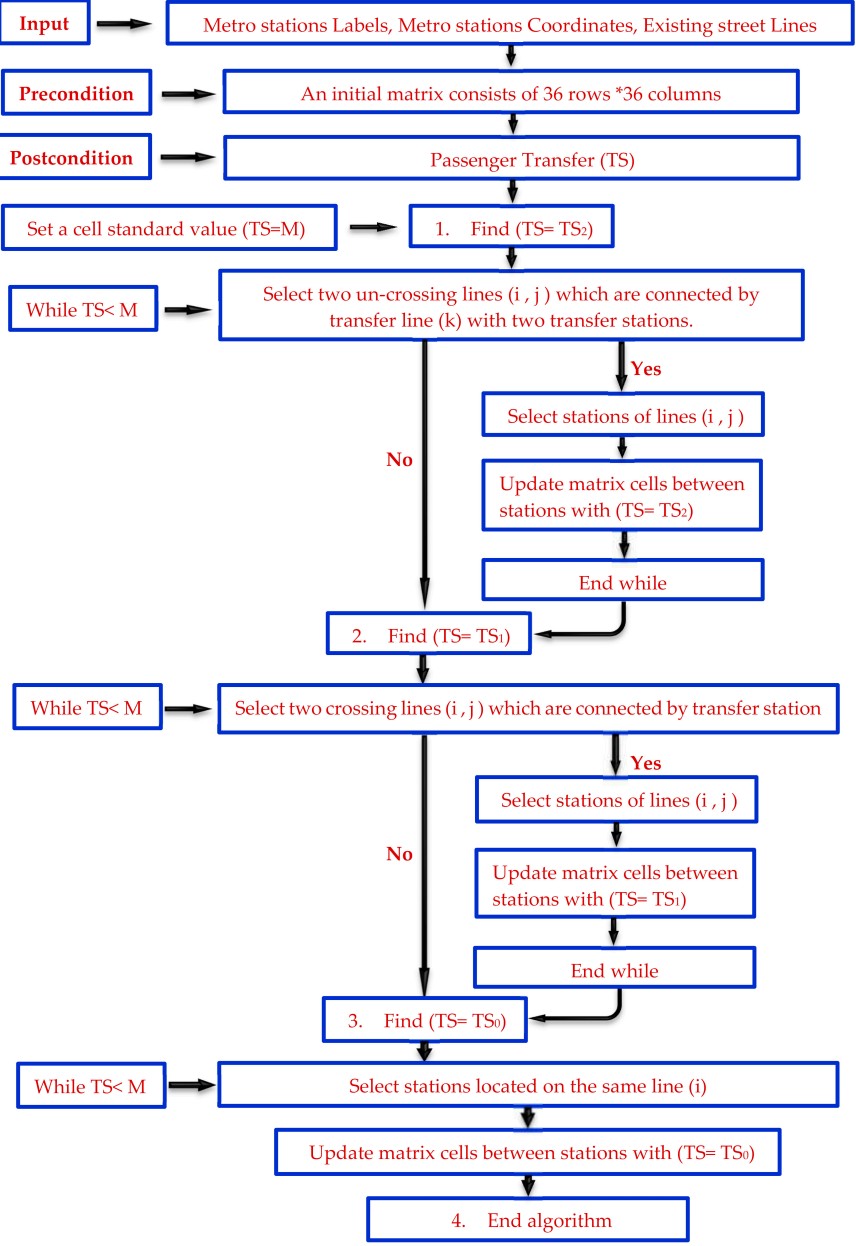

**Figure 7.** Flow chart of TS-algorithm.

### 5.2. Passenger Transfer Matrix of Both Models

In public transit networks, passenger transfer is an important item to evaluate the transit networks that many passengers are usually a concern. The method of calculating the "passenger transfer matrix" was obtained in the paper by calculating the distribution of passenger transfer numbers ($TS_0$, $TS_1$, $TS_2$, $TS_\infty$).

The specific calculation method of the transfer matrix is defined as follows:

- When getting the transfer matrix, the combination of stops in one line is set to 0 if passengers can reach one another without transfer ($TS_0$).
- For other combinations of stops, if they have common stations, the transfer matrix is set to 1, which means that the number of transfer is 1 ($TS_1$).
- If one line doesn't have common stops with another, we should check whether these stops have common stops with the other lines ($TS_2$) or are unserved ($TS_\infty$).

For Model 1, the previous procedures are now used to obtain the transfer matrix for two cases given (Figures 3 and 4) and to look for passenger transfer numbers from the origin (i) to the destination (j) between all 36 stations.

Tables 1 and 2 show the distribution of passenger transfer numbers for grid metro lines in one and both direction, respectively.

The number of transfers depends on network design, which is inside the scope of this work. This work focuses on minimizing the transfer numbers in a given transit network for both models.

The objective of this research is the development of both models for minimizing the transfer numbers in a transit network. The main challenges of designing metro lines in existing cities are summarized as follows:

- Existing streets were already constructed.
- New proposed metro lines were constructed under existing streets as far as possible.
- The new proposed station must be in the existing street intersection.

In this subsection, we define the mean method, which we will use to obtain the distribution of passenger transfer numbers from the passenger transfer matrix. Table 1 shows that the distribution of passenger transfer was caught between (Ts = 0) without transfer and (Ts = ∞) unserved. Figure 3 is interpreting this distribution that all metro lines were constructed in one direction. This is a limitation, because metro lines are not connected in another direction. Thus, in Figure 4, the six metro lines (L1, L2, … … , L6) were connected in the other direction with six metro lines (L7, L8, … … , L12). Table 2 shows the distribution of passenger transfer was caught between (Ts = 0) without transfer and (Ts = 1) one transfer, for all stations to be served (this is an advantage). Nevertheless, this advantage needs a very high total cost to construct metro lines in both directions, as discussed in subsection C. Thus, construction of one metro line in the other direction is sufficient to connect all six metro lines and make all stations be served, as shown in Figure 8.

Model 2 defines the planning of ring and radial metro lines in existing square cities by connecting the proposed stations in a ring and radial direction to be served as possible, as shown in Figures 5 and 6.

We were able to design a grid metro network in existing square cities by constructing metro lines under existing streets. However, design ring-radial metro lines in existing square cities are considered a more complex task than grid metro lines, because of the difference between the paths of ring-radial lines and existing streets from one hand. Moreover, the difficulty of connecting all proposing stations by ring-radial lines from the other hand. Thus, in Model 2, we propose two cases (Figures 5 and 6) and another case (Figure 9). Therefore, Model 2 gives results that are at least as good (and probably better) values for passenger transfer numbers between stations. Table 3 shows that the distribution of passenger transfer was caught between (Ts = 0) without transfer, (Ts = ∞) unserved, (Ts = 1) one transfer, and (Ts = 2) two transfer.

**Table 1.** Passenger Transfer OD Matrix of grid metro lines in one direction (Case 1).

| Station | 1 | 2 | 3 | 4 | 5 | 6 | 7 | 8 | 9 | 10 | 11 | 12 | 13 | 14 | 15 | 16 | 17 | 18 | 19 | 20 | 21 | 22 | 23 | 24 | 25 | 26 | 27 | 28 | 29 | 30 | 31 | 32 | 33 | 34 | 35 | 36 |
|---|---|---|---|---|---|---|---|---|---|---|---|---|---|---|---|---|---|---|---|---|---|---|---|---|---|---|---|---|---|---|---|---|---|---|---|---|
| 1 | 0 | 0 | 0 | 0 | 0 | 0 | ∞ | ∞ | ∞ | ∞ | ∞ | ∞ | ∞ | ∞ | ∞ | ∞ | ∞ | ∞ | ∞ | ∞ | ∞ | ∞ | ∞ | ∞ | ∞ | ∞ | ∞ | ∞ | ∞ | ∞ | ∞ | ∞ | ∞ | ∞ | ∞ | ∞ |
| 2 | 0 | 0 | 0 | 0 | 0 | 0 | ∞ | ∞ | ∞ | ∞ | ∞ | ∞ | ∞ | ∞ | ∞ | ∞ | ∞ | ∞ | ∞ | ∞ | ∞ | ∞ | ∞ | ∞ | ∞ | ∞ | ∞ | ∞ | ∞ | ∞ | ∞ | ∞ | ∞ | ∞ | ∞ | ∞ |
| 3 | 0 | 0 | 0 | 0 | 0 | 0 | ∞ | ∞ | ∞ | ∞ | ∞ | ∞ | ∞ | ∞ | ∞ | ∞ | ∞ | ∞ | ∞ | ∞ | ∞ | ∞ | ∞ | ∞ | ∞ | ∞ | ∞ | ∞ | ∞ | ∞ | ∞ | ∞ | ∞ | ∞ | ∞ | ∞ |
| 4 | 0 | 0 | 0 | 0 | 0 | 0 | ∞ | ∞ | ∞ | ∞ | ∞ | ∞ | ∞ | ∞ | ∞ | ∞ | ∞ | ∞ | ∞ | ∞ | ∞ | ∞ | ∞ | ∞ | ∞ | ∞ | ∞ | ∞ | ∞ | ∞ | ∞ | ∞ | ∞ | ∞ | ∞ | ∞ |
| 5 | 0 | 0 | 0 | 0 | 0 | 0 | ∞ | ∞ | ∞ | ∞ | ∞ | ∞ | ∞ | ∞ | ∞ | ∞ | ∞ | ∞ | ∞ | ∞ | ∞ | ∞ | ∞ | ∞ | ∞ | ∞ | ∞ | ∞ | ∞ | ∞ | ∞ | ∞ | ∞ | ∞ | ∞ | ∞ |
| 6 | 0 | 0 | 0 | 0 | 0 | 0 | ∞ | ∞ | ∞ | ∞ | ∞ | ∞ | ∞ | ∞ | ∞ | ∞ | ∞ | ∞ | ∞ | ∞ | ∞ | ∞ | ∞ | ∞ | ∞ | ∞ | ∞ | ∞ | ∞ | ∞ | ∞ | ∞ | ∞ | ∞ | ∞ | ∞ |
| 7 | ∞ | ∞ | ∞ | ∞ | ∞ | ∞ | 0 | 0 | 0 | 0 | 0 | 0 | ∞ | ∞ | ∞ | ∞ | ∞ | ∞ | ∞ | ∞ | ∞ | ∞ | ∞ | ∞ | ∞ | ∞ | ∞ | ∞ | ∞ | ∞ | ∞ | ∞ | ∞ | ∞ | ∞ | ∞ |
| 8 | ∞ | ∞ | ∞ | ∞ | ∞ | ∞ | 0 | 0 | 0 | 0 | 0 | 0 | ∞ | ∞ | ∞ | ∞ | ∞ | ∞ | ∞ | ∞ | ∞ | ∞ | ∞ | ∞ | ∞ | ∞ | ∞ | ∞ | ∞ | ∞ | ∞ | ∞ | ∞ | ∞ | ∞ | ∞ |
| 9 | ∞ | ∞ | ∞ | ∞ | ∞ | ∞ | 0 | 0 | 0 | 0 | 0 | 0 | ∞ | ∞ | ∞ | ∞ | ∞ | ∞ | ∞ | ∞ | ∞ | ∞ | ∞ | ∞ | ∞ | ∞ | ∞ | ∞ | ∞ | ∞ | ∞ | ∞ | ∞ | ∞ | ∞ | ∞ |
| 10 | ∞ | ∞ | ∞ | ∞ | ∞ | ∞ | 0 | 0 | 0 | 0 | 0 | 0 | ∞ | ∞ | ∞ | ∞ | ∞ | ∞ | ∞ | ∞ | ∞ | ∞ | ∞ | ∞ | ∞ | ∞ | ∞ | ∞ | ∞ | ∞ | ∞ | ∞ | ∞ | ∞ | ∞ | ∞ |
| 11 | ∞ | ∞ | ∞ | ∞ | ∞ | ∞ | 0 | 0 | 0 | 0 | 0 | 0 | ∞ | ∞ | ∞ | ∞ | ∞ | ∞ | ∞ | ∞ | ∞ | ∞ | ∞ | ∞ | ∞ | ∞ | ∞ | ∞ | ∞ | ∞ | ∞ | ∞ | ∞ | ∞ | ∞ | ∞ |
| 12 | ∞ | ∞ | ∞ | ∞ | ∞ | ∞ | 0 | 0 | 0 | 0 | 0 | 0 | ∞ | ∞ | ∞ | ∞ | ∞ | ∞ | ∞ | ∞ | ∞ | ∞ | ∞ | ∞ | ∞ | ∞ | ∞ | ∞ | ∞ | ∞ | ∞ | ∞ | ∞ | ∞ | ∞ | ∞ |
| 13 | ∞ | ∞ | ∞ | ∞ | ∞ | ∞ | ∞ | ∞ | ∞ | ∞ | ∞ | ∞ | 0 | 0 | 0 | 0 | 0 | 0 | ∞ | ∞ | ∞ | ∞ | ∞ | ∞ | ∞ | ∞ | ∞ | ∞ | ∞ | ∞ | ∞ | ∞ | ∞ | ∞ | ∞ | ∞ |
| 14 | ∞ | ∞ | ∞ | ∞ | ∞ | ∞ | ∞ | ∞ | ∞ | ∞ | ∞ | ∞ | 0 | 0 | 0 | 0 | 0 | 0 | ∞ | ∞ | ∞ | ∞ | ∞ | ∞ | ∞ | ∞ | ∞ | ∞ | ∞ | ∞ | ∞ | ∞ | ∞ | ∞ | ∞ | ∞ |
| 15 | ∞ | ∞ | ∞ | ∞ | ∞ | ∞ | ∞ | ∞ | ∞ | ∞ | ∞ | ∞ | 0 | 0 | 0 | 0 | 0 | 0 | ∞ | ∞ | ∞ | ∞ | ∞ | ∞ | ∞ | ∞ | ∞ | ∞ | ∞ | ∞ | ∞ | ∞ | ∞ | ∞ | ∞ | ∞ |
| 16 | ∞ | ∞ | ∞ | ∞ | ∞ | ∞ | ∞ | ∞ | ∞ | ∞ | ∞ | ∞ | 0 | 0 | 0 | 0 | 0 | 0 | ∞ | ∞ | ∞ | ∞ | ∞ | ∞ | ∞ | ∞ | ∞ | ∞ | ∞ | ∞ | ∞ | ∞ | ∞ | ∞ | ∞ | ∞ |
| 17 | ∞ | ∞ | ∞ | ∞ | ∞ | ∞ | ∞ | ∞ | ∞ | ∞ | ∞ | ∞ | 0 | 0 | 0 | 0 | 0 | 0 | ∞ | ∞ | ∞ | ∞ | ∞ | ∞ | ∞ | ∞ | ∞ | ∞ | ∞ | ∞ | ∞ | ∞ | ∞ | ∞ | ∞ | ∞ |
| 18 | ∞ | ∞ | ∞ | ∞ | ∞ | ∞ | ∞ | ∞ | ∞ | ∞ | ∞ | ∞ | 0 | 0 | 0 | 0 | 0 | 0 | ∞ | ∞ | ∞ | ∞ | ∞ | ∞ | ∞ | ∞ | ∞ | ∞ | ∞ | ∞ | ∞ | ∞ | ∞ | ∞ | ∞ | ∞ |
| 19 | ∞ | ∞ | ∞ | ∞ | ∞ | ∞ | ∞ | ∞ | ∞ | ∞ | ∞ | ∞ | ∞ | ∞ | ∞ | ∞ | ∞ | ∞ | 0 | 0 | 0 | 0 | 0 | 0 | ∞ | ∞ | ∞ | ∞ | ∞ | ∞ | ∞ | ∞ | ∞ | ∞ | ∞ | ∞ |
| 20 | ∞ | ∞ | ∞ | ∞ | ∞ | ∞ | ∞ | ∞ | ∞ | ∞ | ∞ | ∞ | ∞ | ∞ | ∞ | ∞ | ∞ | ∞ | 0 | 0 | 0 | 0 | 0 | 0 | ∞ | ∞ | ∞ | ∞ | ∞ | ∞ | ∞ | ∞ | ∞ | ∞ | ∞ | ∞ |
| 21 | ∞ | ∞ | ∞ | ∞ | ∞ | ∞ | ∞ | ∞ | ∞ | ∞ | ∞ | ∞ | ∞ | ∞ | ∞ | ∞ | ∞ | ∞ | 0 | 0 | 0 | 0 | 0 | 0 | ∞ | ∞ | ∞ | ∞ | ∞ | ∞ | ∞ | ∞ | ∞ | ∞ | ∞ | ∞ |
| 22 | ∞ | ∞ | ∞ | ∞ | ∞ | ∞ | ∞ | ∞ | ∞ | ∞ | ∞ | ∞ | ∞ | ∞ | ∞ | ∞ | ∞ | ∞ | 0 | 0 | 0 | 0 | 0 | 0 | ∞ | ∞ | ∞ | ∞ | ∞ | ∞ | ∞ | ∞ | ∞ | ∞ | ∞ | ∞ |
| 23 | ∞ | ∞ | ∞ | ∞ | ∞ | ∞ | ∞ | ∞ | ∞ | ∞ | ∞ | ∞ | ∞ | ∞ | ∞ | ∞ | ∞ | ∞ | 0 | 0 | 0 | 0 | 0 | 0 | ∞ | ∞ | ∞ | ∞ | ∞ | ∞ | ∞ | ∞ | ∞ | ∞ | ∞ | ∞ |
| 24 | ∞ | ∞ | ∞ | ∞ | ∞ | ∞ | ∞ | ∞ | ∞ | ∞ | ∞ | ∞ | ∞ | ∞ | ∞ | ∞ | ∞ | ∞ | 0 | 0 | 0 | 0 | 0 | 0 | ∞ | ∞ | ∞ | ∞ | ∞ | ∞ | ∞ | ∞ | ∞ | ∞ | ∞ | ∞ |
| 25 | ∞ | ∞ | ∞ | ∞ | ∞ | ∞ | ∞ | ∞ | ∞ | ∞ | ∞ | ∞ | ∞ | ∞ | ∞ | ∞ | ∞ | ∞ | ∞ | ∞ | ∞ | ∞ | ∞ | ∞ | 0 | 0 | 0 | 0 | 0 | 0 | ∞ | ∞ | ∞ | ∞ | ∞ | ∞ |
| 26 | ∞ | ∞ | ∞ | ∞ | ∞ | ∞ | ∞ | ∞ | ∞ | ∞ | ∞ | ∞ | ∞ | ∞ | ∞ | ∞ | ∞ | ∞ | ∞ | ∞ | ∞ | ∞ | ∞ | ∞ | 0 | 0 | 0 | 0 | 0 | 0 | ∞ | ∞ | ∞ | ∞ | ∞ | ∞ |
| 27 | ∞ | ∞ | ∞ | ∞ | ∞ | ∞ | ∞ | ∞ | ∞ | ∞ | ∞ | ∞ | ∞ | ∞ | ∞ | ∞ | ∞ | ∞ | ∞ | ∞ | ∞ | ∞ | ∞ | ∞ | 0 | 0 | 0 | 0 | 0 | 0 | ∞ | ∞ | ∞ | ∞ | ∞ | ∞ |
| 28 | ∞ | ∞ | ∞ | ∞ | ∞ | ∞ | ∞ | ∞ | ∞ | ∞ | ∞ | ∞ | ∞ | ∞ | ∞ | ∞ | ∞ | ∞ | ∞ | ∞ | ∞ | ∞ | ∞ | ∞ | 0 | 0 | 0 | 0 | 0 | 0 | ∞ | ∞ | ∞ | ∞ | ∞ | ∞ |
| 29 | ∞ | ∞ | ∞ | ∞ | ∞ | ∞ | ∞ | ∞ | ∞ | ∞ | ∞ | ∞ | ∞ | ∞ | ∞ | ∞ | ∞ | ∞ | ∞ | ∞ | ∞ | ∞ | ∞ | ∞ | 0 | 0 | 0 | 0 | 0 | 0 | ∞ | ∞ | ∞ | ∞ | ∞ | ∞ |
| 30 | ∞ | ∞ | ∞ | ∞ | ∞ | ∞ | ∞ | ∞ | ∞ | ∞ | ∞ | ∞ | ∞ | ∞ | ∞ | ∞ | ∞ | ∞ | ∞ | ∞ | ∞ | ∞ | ∞ | ∞ | 0 | 0 | 0 | 0 | 0 | 0 | ∞ | ∞ | ∞ | ∞ | ∞ | ∞ |
| 31 | ∞ | ∞ | ∞ | ∞ | ∞ | ∞ | ∞ | ∞ | ∞ | ∞ | ∞ | ∞ | ∞ | ∞ | ∞ | ∞ | ∞ | ∞ | ∞ | ∞ | ∞ | ∞ | ∞ | ∞ | ∞ | ∞ | ∞ | ∞ | ∞ | ∞ | 0 | 0 | 0 | 0 | 0 | 0 |
| 32 | ∞ | ∞ | ∞ | ∞ | ∞ | ∞ | ∞ | ∞ | ∞ | ∞ | ∞ | ∞ | ∞ | ∞ | ∞ | ∞ | ∞ | ∞ | ∞ | ∞ | ∞ | ∞ | ∞ | ∞ | ∞ | ∞ | ∞ | ∞ | ∞ | ∞ | 0 | 0 | 0 | 0 | 0 | 0 |
| 33 | ∞ | ∞ | ∞ | ∞ | ∞ | ∞ | ∞ | ∞ | ∞ | ∞ | ∞ | ∞ | ∞ | ∞ | ∞ | ∞ | ∞ | ∞ | ∞ | ∞ | ∞ | ∞ | ∞ | ∞ | ∞ | ∞ | ∞ | ∞ | ∞ | ∞ | 0 | 0 | 0 | 0 | 0 | 0 |
| 34 | ∞ | ∞ | ∞ | ∞ | ∞ | ∞ | ∞ | ∞ | ∞ | ∞ | ∞ | ∞ | ∞ | ∞ | ∞ | ∞ | ∞ | ∞ | ∞ | ∞ | ∞ | ∞ | ∞ | ∞ | ∞ | ∞ | ∞ | ∞ | ∞ | ∞ | 0 | 0 | 0 | 0 | 0 | 0 |
| 35 | ∞ | ∞ | ∞ | ∞ | ∞ | ∞ | ∞ | ∞ | ∞ | ∞ | ∞ | ∞ | ∞ | ∞ | ∞ | ∞ | ∞ | ∞ | ∞ | ∞ | ∞ | ∞ | ∞ | ∞ | ∞ | ∞ | ∞ | ∞ | ∞ | ∞ | 0 | 0 | 0 | 0 | 0 | 0 |
| 36 | ∞ | ∞ | ∞ | ∞ | ∞ | ∞ | ∞ | ∞ | ∞ | ∞ | ∞ | ∞ | ∞ | ∞ | ∞ | ∞ | ∞ | ∞ | ∞ | ∞ | ∞ | ∞ | ∞ | ∞ | ∞ | ∞ | ∞ | ∞ | ∞ | ∞ | 0 | 0 | 0 | 0 | 0 | 0 |

**Table 2.** Passenger Transfer OD Matrix of grid metro lines in both directions (Case 2).

| Station | 1 | 2 | 3 | 4 | 5 | 6 | 7 | 8 | 9 | 10 | 11 | 12 | 13 | 14 | 15 | 16 | 17 | 18 | 19 | 20 | 21 | 22 | 23 | 24 | 25 | 26 | 27 | 28 | 29 | 30 | 31 | 32 | 33 | 34 | 35 | 36 |
|---|---|---|---|---|---|---|---|---|---|---|---|---|---|---|---|---|---|---|---|---|---|---|---|---|---|---|---|---|---|---|---|---|---|---|---|---|
| 1 | 0 | 0 | 0 | 0 | 0 | 0 | 0 | 1 | 1 | 1 | 1 | 1 | 0 | 1 | 1 | 1 | 1 | 1 | 0 | 1 | 1 | 1 | 1 | 1 | 0 | 1 | 1 | 1 | 1 | 1 | 0 | 1 | 1 | 1 | 1 | 1 |
| 2 | 0 | 0 | 0 | 0 | 0 | 0 | 1 | 0 | 1 | 1 | 1 | 1 | 1 | 0 | 1 | 1 | 1 | 1 | 1 | 0 | 1 | 1 | 1 | 1 | 1 | 0 | 1 | 1 | 1 | 1 | 1 | 0 | 1 | 1 | 1 | 1 |
| 3 | 0 | 0 | 0 | 0 | 0 | 0 | 1 | 1 | 0 | 1 | 1 | 1 | 1 | 1 | 0 | 1 | 1 | 1 | 1 | 1 | 0 | 1 | 1 | 1 | 1 | 1 | 0 | 1 | 1 | 1 | 1 | 1 | 0 | 1 | 1 | 1 |
| 4 | 0 | 0 | 0 | 0 | 0 | 0 | 1 | 1 | 1 | 0 | 1 | 1 | 1 | 1 | 1 | 0 | 1 | 1 | 1 | 1 | 1 | 0 | 1 | 1 | 1 | 1 | 1 | 0 | 1 | 1 | 1 | 1 | 1 | 0 | 1 | 1 |
| 5 | 0 | 0 | 0 | 0 | 0 | 0 | 1 | 1 | 1 | 1 | 0 | 1 | 1 | 1 | 1 | 1 | 0 | 1 | 1 | 1 | 1 | 1 | 0 | 1 | 1 | 1 | 1 | 1 | 0 | 1 | 1 | 1 | 1 | 1 | 0 | 1 |
| 6 | 0 | 0 | 0 | 0 | 0 | 0 | 1 | 1 | 1 | 1 | 1 | 0 | 1 | 1 | 1 | 1 | 1 | 0 | 1 | 1 | 1 | 1 | 1 | 0 | 1 | 1 | 1 | 1 | 1 | 0 | 1 | 1 | 1 | 1 | 1 | 0 |
| 7 | 0 | 1 | 1 | 1 | 1 | 1 | 0 | 0 | 0 | 0 | 0 | 0 | 0 | 1 | 1 | 1 | 1 | 1 | 0 | 1 | 1 | 1 | 1 | 1 | 0 | 1 | 1 | 1 | 1 | 1 | 0 | 1 | 1 | 1 | 1 | 1 |
| 8 | 1 | 0 | 1 | 1 | 1 | 1 | 0 | 0 | 0 | 0 | 0 | 0 | 1 | 0 | 1 | 1 | 1 | 1 | 1 | 0 | 1 | 1 | 1 | 1 | 1 | 0 | 1 | 1 | 1 | 1 | 1 | 0 | 1 | 1 | 1 | 1 |
| 9 | 1 | 1 | 0 | 1 | 1 | 1 | 0 | 0 | 0 | 0 | 0 | 0 | 1 | 1 | 0 | 1 | 1 | 1 | 1 | 1 | 0 | 1 | 1 | 1 | 1 | 1 | 0 | 1 | 1 | 1 | 1 | 1 | 0 | 1 | 1 | 1 |
| 10 | 1 | 1 | 1 | 0 | 1 | 1 | 0 | 0 | 0 | 0 | 0 | 0 | 1 | 1 | 1 | 0 | 1 | 1 | 1 | 1 | 1 | 0 | 1 | 1 | 1 | 1 | 1 | 0 | 1 | 1 | 1 | 1 | 1 | 0 | 1 | 1 |
| 11 | 1 | 1 | 1 | 1 | 0 | 1 | 0 | 0 | 0 | 0 | 0 | 0 | 1 | 1 | 1 | 1 | 0 | 1 | 1 | 1 | 1 | 1 | 0 | 1 | 1 | 1 | 1 | 1 | 0 | 1 | 1 | 1 | 1 | 1 | 0 | 1 |
| 12 | 1 | 1 | 1 | 1 | 1 | 0 | 0 | 0 | 0 | 0 | 0 | 0 | 1 | 1 | 1 | 1 | 1 | 0 | 1 | 1 | 1 | 1 | 1 | 0 | 1 | 1 | 1 | 1 | 1 | 0 | 1 | 1 | 1 | 1 | 1 | 0 |
| 13 | 0 | 1 | 1 | 1 | 1 | 1 | 0 | 1 | 1 | 1 | 1 | 1 | 0 | 0 | 0 | 0 | 0 | 0 | 0 | 1 | 1 | 1 | 1 | 1 | 0 | 1 | 1 | 1 | 1 | 1 | 0 | 1 | 1 | 1 | 1 | 1 |
| 14 | 1 | 0 | 1 | 1 | 1 | 1 | 1 | 0 | 1 | 1 | 1 | 1 | 0 | 0 | 0 | 0 | 0 | 0 | 1 | 0 | 1 | 1 | 1 | 1 | 1 | 0 | 1 | 1 | 1 | 1 | 1 | 0 | 1 | 1 | 1 | 1 |
| 15 | 1 | 1 | 0 | 1 | 1 | 1 | 1 | 1 | 0 | 1 | 1 | 1 | 0 | 0 | 0 | 0 | 0 | 0 | 1 | 1 | 0 | 1 | 1 | 1 | 1 | 1 | 0 | 1 | 1 | 1 | 1 | 1 | 0 | 1 | 1 | 1 |
| 16 | 1 | 1 | 1 | 0 | 1 | 1 | 1 | 1 | 1 | 0 | 1 | 1 | 0 | 0 | 0 | 0 | 0 | 0 | 1 | 1 | 1 | 0 | 1 | 1 | 1 | 1 | 1 | 0 | 1 | 1 | 1 | 1 | 1 | 0 | 1 | 1 |
| 17 | 1 | 1 | 1 | 1 | 0 | 1 | 1 | 1 | 1 | 1 | 0 | 1 | 0 | 0 | 0 | 0 | 0 | 0 | 1 | 1 | 1 | 1 | 0 | 1 | 1 | 1 | 1 | 1 | 0 | 1 | 1 | 1 | 1 | 1 | 0 | 1 |
| 18 | 1 | 1 | 1 | 1 | 1 | 0 | 1 | 1 | 1 | 1 | 1 | 0 | 0 | 0 | 0 | 0 | 0 | 0 | 1 | 1 | 1 | 1 | 1 | 0 | 1 | 1 | 1 | 1 | 1 | 0 | 1 | 1 | 1 | 1 | 1 | 0 |
| 19 | 0 | 1 | 1 | 1 | 1 | 1 | 0 | 1 | 1 | 1 | 1 | 1 | 0 | 1 | 1 | 1 | 1 | 1 | 0 | 0 | 0 | 0 | 0 | 0 | 0 | 1 | 1 | 1 | 1 | 1 | 0 | 1 | 1 | 1 | 1 | 1 |
| 20 | 1 | 0 | 1 | 1 | 1 | 1 | 1 | 0 | 1 | 1 | 1 | 1 | 1 | 0 | 1 | 1 | 1 | 1 | 0 | 0 | 0 | 0 | 0 | 0 | 1 | 0 | 1 | 1 | 1 | 1 | 1 | 0 | 1 | 1 | 1 | 1 |
| 21 | 1 | 1 | 0 | 1 | 1 | 1 | 1 | 1 | 0 | 1 | 1 | 1 | 1 | 1 | 0 | 1 | 1 | 1 | 0 | 0 | 0 | 0 | 0 | 0 | 1 | 1 | 0 | 1 | 1 | 1 | 1 | 1 | 0 | 1 | 1 | 1 |
| 22 | 1 | 1 | 1 | 0 | 1 | 1 | 1 | 1 | 1 | 0 | 1 | 1 | 1 | 1 | 1 | 0 | 1 | 1 | 0 | 0 | 0 | 0 | 0 | 0 | 1 | 1 | 1 | 0 | 1 | 1 | 1 | 1 | 1 | 0 | 1 | 1 |
| 23 | 1 | 1 | 1 | 1 | 0 | 1 | 1 | 1 | 1 | 1 | 0 | 1 | 1 | 1 | 1 | 1 | 0 | 1 | 0 | 0 | 0 | 0 | 0 | 0 | 1 | 1 | 1 | 1 | 0 | 1 | 1 | 1 | 1 | 1 | 0 | 1 |
| 24 | 1 | 1 | 1 | 1 | 1 | 0 | 1 | 1 | 1 | 1 | 1 | 0 | 1 | 1 | 1 | 1 | 1 | 0 | 0 | 0 | 0 | 0 | 0 | 0 | 1 | 1 | 1 | 1 | 1 | 0 | 1 | 1 | 1 | 1 | 1 | 0 |
| 25 | 0 | 1 | 1 | 1 | 1 | 1 | 0 | 1 | 1 | 1 | 1 | 1 | 0 | 1 | 1 | 1 | 1 | 1 | 0 | 1 | 1 | 1 | 1 | 1 | 0 | 0 | 0 | 0 | 0 | 0 | 0 | 1 | 1 | 1 | 1 | 1 |
| 26 | 1 | 0 | 1 | 1 | 1 | 1 | 1 | 0 | 1 | 1 | 1 | 1 | 1 | 0 | 1 | 1 | 1 | 1 | 1 | 0 | 1 | 1 | 1 | 1 | 0 | 0 | 0 | 0 | 0 | 0 | 1 | 0 | 1 | 1 | 1 | 1 |
| 27 | 1 | 1 | 0 | 1 | 1 | 1 | 1 | 1 | 0 | 1 | 1 | 1 | 1 | 1 | 0 | 1 | 1 | 1 | 1 | 1 | 0 | 1 | 1 | 1 | 0 | 0 | 0 | 0 | 0 | 0 | 1 | 1 | 0 | 1 | 1 | 1 |
| 28 | 1 | 1 | 1 | 0 | 1 | 1 | 1 | 1 | 1 | 0 | 1 | 1 | 1 | 1 | 1 | 0 | 1 | 1 | 1 | 1 | 1 | 0 | 1 | 1 | 0 | 0 | 0 | 0 | 0 | 0 | 1 | 1 | 1 | 0 | 1 | 1 |
| 29 | 1 | 1 | 1 | 1 | 0 | 1 | 1 | 1 | 1 | 1 | 0 | 1 | 1 | 1 | 1 | 1 | 0 | 1 | 1 | 1 | 1 | 1 | 0 | 1 | 0 | 0 | 0 | 0 | 0 | 0 | 1 | 1 | 1 | 1 | 0 | 1 |
| 30 | 1 | 1 | 1 | 1 | 1 | 0 | 1 | 1 | 1 | 1 | 1 | 0 | 1 | 1 | 1 | 1 | 1 | 0 | 1 | 1 | 1 | 1 | 1 | 0 | 0 | 0 | 0 | 0 | 0 | 0 | 1 | 1 | 1 | 1 | 1 | 0 |
| 31 | 0 | 1 | 1 | 1 | 1 | 1 | 0 | 1 | 1 | 1 | 1 | 1 | 0 | 1 | 1 | 1 | 1 | 1 | 0 | 1 | 1 | 1 | 1 | 1 | 0 | 1 | 1 | 1 | 1 | 1 | 0 | 0 | 0 | 0 | 0 | 0 |
| 32 | 1 | 0 | 1 | 1 | 1 | 1 | 1 | 0 | 1 | 1 | 1 | 1 | 1 | 0 | 1 | 1 | 1 | 1 | 1 | 0 | 1 | 1 | 1 | 1 | 1 | 0 | 1 | 1 | 1 | 1 | 0 | 0 | 0 | 0 | 0 | 0 |
| 33 | 1 | 1 | 0 | 1 | 1 | 1 | 1 | 1 | 0 | 1 | 1 | 1 | 1 | 1 | 0 | 1 | 1 | 1 | 1 | 1 | 0 | 1 | 1 | 1 | 1 | 1 | 0 | 1 | 1 | 1 | 0 | 0 | 0 | 0 | 0 | 0 |
| 34 | 1 | 1 | 1 | 0 | 1 | 1 | 1 | 1 | 1 | 0 | 1 | 1 | 1 | 1 | 1 | 0 | 1 | 1 | 1 | 1 | 1 | 0 | 1 | 1 | 1 | 1 | 1 | 0 | 1 | 1 | 0 | 0 | 0 | 0 | 0 | 0 |
| 35 | 1 | 1 | 1 | 1 | 0 | 1 | 1 | 1 | 1 | 1 | 0 | 1 | 1 | 1 | 1 | 1 | 0 | 1 | 1 | 1 | 1 | 1 | 0 | 1 | 1 | 1 | 1 | 1 | 0 | 1 | 0 | 0 | 0 | 0 | 0 | 0 |
| 36 | 1 | 1 | 1 | 1 | 1 | 0 | 1 | 1 | 1 | 1 | 1 | 0 | 1 | 1 | 1 | 1 | 1 | 0 | 1 | 1 | 1 | 1 | 1 | 0 | 1 | 1 | 1 | 1 | 1 | 0 | 0 | 0 | 0 | 0 | 0 | 0 |

**Table 3.** Passenger Transfer OD Matrix of the ring-radial metro lines (Case 3).

| Station | 1 | 2 | 3 | 4 | 5 | 6 | 7 | 8 | 9 | 10 | 11 | 12 | 13 | 14 | 15 | 16 | 17 | 18 | 19 | 20 | 21 | 22 | 23 | 24 | 25 | 26 | 27 | 28 | 29 | 30 | 31 | 32 | 33 | 34 | 35 | 36 |
|---|---|---|---|---|---|---|---|---|---|---|---|---|---|---|---|---|---|---|---|---|---|---|---|---|---|---|---|---|---|---|---|---|---|---|---|---|
| 1 | 0 | ∞ | ∞ | ∞ | ∞ | 2 | ∞ | 0 | ∞ | ∞ | 2 | ∞ | ∞ | ∞ | 0 | 1 | ∞ | ∞ | ∞ | ∞ | 1 | 0 | ∞ | ∞ | ∞ | 2 | ∞ | ∞ | 0 | ∞ | 2 | ∞ | ∞ | ∞ | ∞ | 0 |
| 2 | ∞ | 0 | ∞ | ∞ | ∞ | ∞ | ∞ | ∞ | ∞ | ∞ | ∞ | ∞ | ∞ | ∞ | ∞ | ∞ | ∞ | ∞ | ∞ | ∞ | ∞ | ∞ | ∞ | ∞ | ∞ | ∞ | ∞ | ∞ | ∞ | ∞ | ∞ | ∞ | ∞ | ∞ | ∞ | ∞ |
| 3 | ∞ | ∞ | 0 | 0 | ∞ | ∞ | ∞ | ∞ | ∞ | ∞ | ∞ | ∞ | 0 | ∞ | ∞ | ∞ | ∞ | 0 | 0 | ∞ | ∞ | ∞ | ∞ | 0 | ∞ | ∞ | ∞ | ∞ | ∞ | ∞ | ∞ | ∞ | 0 | 0 | ∞ | ∞ |
| 4 | ∞ | ∞ | 0 | 0 | ∞ | ∞ | ∞ | ∞ | ∞ | ∞ | ∞ | ∞ | 0 | ∞ | ∞ | ∞ | ∞ | 0 | 0 | ∞ | ∞ | ∞ | ∞ | 0 | ∞ | ∞ | ∞ | ∞ | ∞ | ∞ | ∞ | ∞ | 0 | 0 | ∞ | ∞ |
| 5 | ∞ | ∞ | ∞ | ∞ | 0 | ∞ | ∞ | ∞ | ∞ | ∞ | ∞ | ∞ | ∞ | ∞ | ∞ | ∞ | ∞ | ∞ | ∞ | ∞ | ∞ | ∞ | ∞ | ∞ | ∞ | ∞ | ∞ | ∞ | ∞ | ∞ | ∞ | ∞ | ∞ | ∞ | ∞ | ∞ |
| 6 | 2 | ∞ | ∞ | ∞ | ∞ | 0 | ∞ | 2 | ∞ | ∞ | 0 | ∞ | ∞ | ∞ | 1 | 0 | ∞ | ∞ | ∞ | ∞ | 0 | 1 | ∞ | ∞ | ∞ | 0 | ∞ | ∞ | 2 | ∞ | 0 | ∞ | ∞ | ∞ | ∞ | 2 |
| 7 | ∞ | ∞ | ∞ | ∞ | ∞ | ∞ | 0 | ∞ | ∞ | ∞ | ∞ | ∞ | ∞ | ∞ | ∞ | ∞ | ∞ | ∞ | ∞ | ∞ | ∞ | ∞ | ∞ | ∞ | ∞ | ∞ | ∞ | ∞ | ∞ | ∞ | ∞ | ∞ | ∞ | ∞ | ∞ | ∞ |
| 8 | 0 | ∞ | ∞ | ∞ | ∞ | 2 | ∞ | 0 | ∞ | ∞ | 2 | ∞ | ∞ | ∞ | 0 | 1 | ∞ | ∞ | ∞ | ∞ | 1 | 0 | ∞ | ∞ | ∞ | 2 | ∞ | ∞ | 0 | ∞ | 2 | ∞ | ∞ | ∞ | ∞ | 0 |
| 9 | ∞ | ∞ | ∞ | ∞ | ∞ | ∞ | ∞ | ∞ | 0 | 0 | ∞ | ∞ | ∞ | 0 | ∞ | ∞ | 0 | ∞ | ∞ | 0 | ∞ | ∞ | 0 | ∞ | ∞ | ∞ | 0 | ∞ | ∞ | ∞ | ∞ | ∞ | ∞ | ∞ | ∞ | ∞ |
| 10 | ∞ | ∞ | ∞ | ∞ | ∞ | ∞ | ∞ | ∞ | 0 | 0 | ∞ | ∞ | ∞ | 0 | ∞ | ∞ | 0 | ∞ | ∞ | 0 | ∞ | ∞ | 0 | ∞ | ∞ | ∞ | 0 | ∞ | ∞ | ∞ | ∞ | ∞ | ∞ | ∞ | ∞ | ∞ |
| 11 | 2 | ∞ | ∞ | ∞ | ∞ | 0 | ∞ | 2 | ∞ | ∞ | 0 | ∞ | ∞ | ∞ | 1 | 0 | ∞ | ∞ | ∞ | ∞ | 0 | 1 | ∞ | ∞ | ∞ | 0 | ∞ | ∞ | 2 | ∞ | 0 | ∞ | ∞ | ∞ | ∞ | 2 |
| 12 | ∞ | ∞ | ∞ | ∞ | ∞ | ∞ | ∞ | ∞ | ∞ | ∞ | ∞ | 0 | ∞ | ∞ | ∞ | ∞ | ∞ | ∞ | ∞ | ∞ | ∞ | ∞ | ∞ | ∞ | ∞ | ∞ | ∞ | ∞ | ∞ | ∞ | ∞ | ∞ | ∞ | ∞ | ∞ | ∞ |
| 13 | ∞ | ∞ | 0 | 0 | ∞ | ∞ | ∞ | ∞ | ∞ | ∞ | ∞ | ∞ | 0 | ∞ | ∞ | ∞ | ∞ | 0 | 0 | ∞ | ∞ | ∞ | ∞ | 0 | ∞ | ∞ | ∞ | ∞ | ∞ | ∞ | ∞ | ∞ | 0 | 0 | ∞ | ∞ |
| 14 | ∞ | ∞ | ∞ | ∞ | ∞ | ∞ | ∞ | ∞ | 0 | 0 | ∞ | ∞ | ∞ | 0 | ∞ | ∞ | 0 | ∞ | ∞ | 0 | ∞ | ∞ | 0 | ∞ | ∞ | ∞ | 0 | ∞ | ∞ | ∞ | ∞ | ∞ | ∞ | ∞ | ∞ | ∞ |
| 15 | 0 | ∞ | ∞ | ∞ | ∞ | 1 | ∞ | 0 | ∞ | ∞ | 1 | ∞ | ∞ | ∞ | 0 | 0 | ∞ | ∞ | ∞ | ∞ | 0 | 0 | ∞ | ∞ | ∞ | 1 | ∞ | ∞ | 0 | ∞ | 1 | ∞ | ∞ | ∞ | ∞ | 0 |
| 16 | 1 | ∞ | ∞ | ∞ | ∞ | 0 | ∞ | 1 | ∞ | ∞ | 0 | ∞ | ∞ | ∞ | 0 | 0 | ∞ | ∞ | ∞ | ∞ | 0 | 0 | ∞ | ∞ | ∞ | 0 | ∞ | ∞ | 1 | ∞ | 0 | ∞ | ∞ | ∞ | ∞ | 1 |
| 17 | ∞ | ∞ | ∞ | ∞ | ∞ | ∞ | ∞ | ∞ | 0 | 0 | ∞ | ∞ | ∞ | 0 | ∞ | ∞ | 0 | ∞ | ∞ | 0 | ∞ | ∞ | 0 | ∞ | ∞ | ∞ | 0 | ∞ | ∞ | ∞ | ∞ | ∞ | ∞ | ∞ | ∞ | ∞ |
| 18 | ∞ | ∞ | 0 | 0 | ∞ | ∞ | ∞ | ∞ | ∞ | ∞ | ∞ | ∞ | 0 | ∞ | ∞ | ∞ | ∞ | 0 | 0 | ∞ | ∞ | ∞ | ∞ | 0 | ∞ | ∞ | ∞ | ∞ | ∞ | ∞ | ∞ | ∞ | 0 | 0 | ∞ | ∞ |
| 19 | ∞ | ∞ | 0 | 0 | ∞ | ∞ | ∞ | ∞ | ∞ | ∞ | ∞ | ∞ | 0 | ∞ | ∞ | ∞ | ∞ | 0 | 0 | ∞ | ∞ | ∞ | ∞ | 0 | ∞ | ∞ | ∞ | ∞ | ∞ | ∞ | ∞ | ∞ | 0 | 0 | ∞ | ∞ |
| 20 | ∞ | ∞ | ∞ | ∞ | ∞ | ∞ | ∞ | ∞ | 0 | 0 | ∞ | ∞ | ∞ | 0 | ∞ | ∞ | 0 | ∞ | ∞ | 0 | ∞ | ∞ | 0 | ∞ | ∞ | ∞ | 0 | ∞ | ∞ | ∞ | ∞ | ∞ | ∞ | ∞ | ∞ | ∞ |
| 21 | 1 | ∞ | ∞ | ∞ | ∞ | 0 | ∞ | 1 | ∞ | ∞ | 0 | ∞ | ∞ | ∞ | 0 | 0 | ∞ | ∞ | ∞ | ∞ | 0 | 0 | ∞ | ∞ | ∞ | 0 | ∞ | ∞ | 1 | ∞ | 0 | ∞ | ∞ | ∞ | ∞ | 1 |
| 22 | 0 | ∞ | ∞ | ∞ | ∞ | 1 | ∞ | 0 | ∞ | ∞ | 1 | ∞ | ∞ | ∞ | 0 | 0 | ∞ | ∞ | ∞ | ∞ | 0 | 0 | ∞ | ∞ | ∞ | 1 | ∞ | ∞ | 0 | ∞ | 1 | ∞ | ∞ | ∞ | ∞ | 0 |
| 23 | ∞ | ∞ | ∞ | ∞ | ∞ | ∞ | ∞ | ∞ | 0 | 0 | ∞ | ∞ | ∞ | 0 | ∞ | ∞ | 0 | ∞ | ∞ | 0 | ∞ | ∞ | 0 | ∞ | ∞ | ∞ | 0 | ∞ | ∞ | ∞ | ∞ | ∞ | ∞ | ∞ | ∞ | ∞ |
| 24 | ∞ | ∞ | 0 | 0 | ∞ | ∞ | ∞ | ∞ | ∞ | ∞ | ∞ | ∞ | 0 | ∞ | ∞ | ∞ | ∞ | 0 | 0 | ∞ | ∞ | ∞ | ∞ | 0 | ∞ | ∞ | ∞ | ∞ | ∞ | ∞ | ∞ | ∞ | 0 | 0 | ∞ | ∞ |
| 25 | ∞ | ∞ | ∞ | ∞ | ∞ | ∞ | ∞ | ∞ | ∞ | ∞ | ∞ | ∞ | ∞ | ∞ | ∞ | ∞ | ∞ | ∞ | ∞ | ∞ | ∞ | ∞ | ∞ | ∞ | 0 | ∞ | ∞ | ∞ | ∞ | ∞ | ∞ | ∞ | ∞ | ∞ | ∞ | ∞ |
| 26 | 2 | ∞ | ∞ | ∞ | ∞ | 0 | ∞ | 2 | ∞ | ∞ | 0 | ∞ | ∞ | ∞ | 1 | 0 | ∞ | ∞ | ∞ | ∞ | 0 | 1 | ∞ | ∞ | ∞ | 0 | ∞ | ∞ | 2 | ∞ | 0 | ∞ | ∞ | ∞ | ∞ | 2 |
| 27 | ∞ | ∞ | ∞ | ∞ | ∞ | ∞ | ∞ | ∞ | 0 | 0 | ∞ | ∞ | ∞ | 0 | ∞ | ∞ | 0 | ∞ | ∞ | 0 | ∞ | ∞ | 0 | ∞ | ∞ | ∞ | 0 | ∞ | ∞ | ∞ | ∞ | ∞ | ∞ | ∞ | ∞ | ∞ |
| 28 | ∞ | ∞ | ∞ | ∞ | ∞ | ∞ | ∞ | ∞ | 0 | 0 | ∞ | ∞ | ∞ | 0 | ∞ | ∞ | 0 | ∞ | ∞ | 0 | ∞ | ∞ | 0 | ∞ | ∞ | ∞ | 0 | ∞ | ∞ | ∞ | ∞ | ∞ | ∞ | ∞ | ∞ | ∞ |
| 29 | 0 | ∞ | ∞ | ∞ | ∞ | 2 | ∞ | 0 | ∞ | ∞ | 2 | ∞ | ∞ | ∞ | 0 | 1 | ∞ | ∞ | ∞ | ∞ | 1 | 0 | ∞ | ∞ | ∞ | 2 | ∞ | ∞ | 0 | ∞ | 2 | ∞ | ∞ | ∞ | ∞ | 0 |
| 30 | ∞ | ∞ | ∞ | ∞ | ∞ | ∞ | ∞ | ∞ | ∞ | ∞ | ∞ | ∞ | ∞ | ∞ | ∞ | ∞ | ∞ | ∞ | ∞ | ∞ | ∞ | ∞ | ∞ | ∞ | ∞ | ∞ | ∞ | ∞ | ∞ | 0 | ∞ | ∞ | ∞ | ∞ | ∞ | ∞ |
| 31 | 2 | ∞ | ∞ | ∞ | ∞ | 0 | ∞ | 2 | ∞ | ∞ | 0 | ∞ | ∞ | ∞ | 1 | 0 | ∞ | ∞ | ∞ | ∞ | 0 | 1 | ∞ | ∞ | ∞ | 0 | ∞ | ∞ | 2 | ∞ | 0 | ∞ | ∞ | ∞ | ∞ | 2 |
| 32 | ∞ | ∞ | ∞ | ∞ | ∞ | ∞ | ∞ | ∞ | ∞ | ∞ | ∞ | ∞ | ∞ | ∞ | ∞ | ∞ | ∞ | ∞ | ∞ | ∞ | ∞ | ∞ | ∞ | ∞ | ∞ | ∞ | ∞ | ∞ | ∞ | ∞ | ∞ | 0 | ∞ | ∞ | ∞ | ∞ |
| 33 | ∞ | ∞ | 0 | 0 | ∞ | ∞ | ∞ | ∞ | ∞ | ∞ | ∞ | ∞ | 0 | ∞ | ∞ | ∞ | ∞ | 0 | 0 | ∞ | ∞ | ∞ | ∞ | 0 | ∞ | ∞ | ∞ | ∞ | ∞ | ∞ | ∞ | ∞ | 0 | 0 | ∞ | ∞ |
| 34 | ∞ | ∞ | 0 | 0 | ∞ | ∞ | ∞ | ∞ | ∞ | ∞ | ∞ | ∞ | 0 | ∞ | ∞ | ∞ | ∞ | 0 | 0 | ∞ | ∞ | ∞ | ∞ | 0 | ∞ | ∞ | ∞ | ∞ | ∞ | ∞ | ∞ | ∞ | 0 | 0 | ∞ | ∞ |
| 35 | ∞ | ∞ | ∞ | ∞ | ∞ | ∞ | ∞ | ∞ | ∞ | ∞ | ∞ | ∞ | ∞ | ∞ | ∞ | ∞ | ∞ | ∞ | ∞ | ∞ | ∞ | ∞ | ∞ | ∞ | ∞ | ∞ | ∞ | ∞ | ∞ | ∞ | ∞ | ∞ | ∞ | ∞ | 0 | ∞ |
| 36 | 0 | ∞ | ∞ | ∞ | ∞ | 2 | ∞ | 0 | ∞ | ∞ | 2 | ∞ | ∞ | ∞ | 0 | 1 | ∞ | ∞ | ∞ | ∞ | 1 | 0 | ∞ | ∞ | ∞ | 2 | ∞ | ∞ | 0 | ∞ | 2 | ∞ | ∞ | ∞ | ∞ | 0 |

In Figure 5, the interpretation of this distribution (most stations are not served) is that the connection of ring and radial metro lines are not transferred stations (because the connections point is not at the intersection of the existing streets). Furthermore, some proposed stations are not connected with metro lines. This is a limitation because metro lines are not connected, and most of the stations are not served. Thus, in Figure 6, there is a modification in the path of metro lines (L3, L4, and L5), so the proposing station are connected with the metro line, and the connection of ring-radial lines is a transfer station if possible. Table 4 shows that the distribution of passenger transfer was caught between, (Ts = 0) without transfer, (Ts = ∞) unserved, (Ts = 1) one transfer, and (Ts = 2) two transfer, so most of the stations are served (this is an advantage). However, this advantage needs a very high total cost to construct two radial lines, as discussed in subsection C. Thus, construction of one metro radial line is sufficient to connect most metro lines and make most of the stations to be served, as shown in Figure 9.

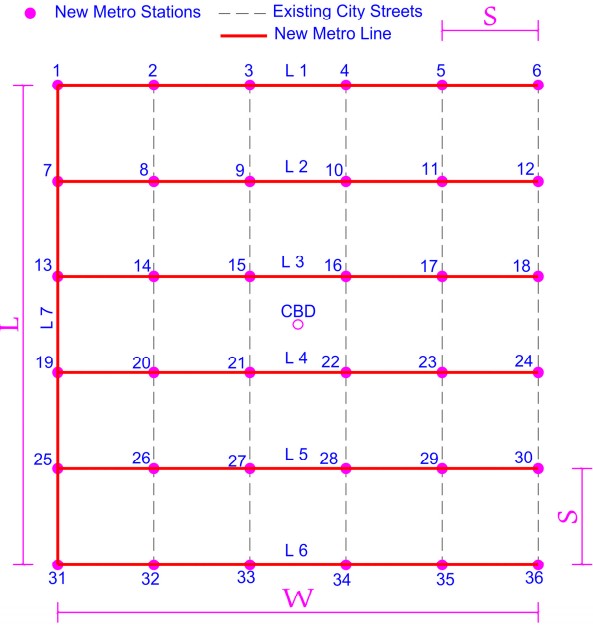

**Figure 8.** Grid metro lines are connected with one line in the other directions (Case 5).

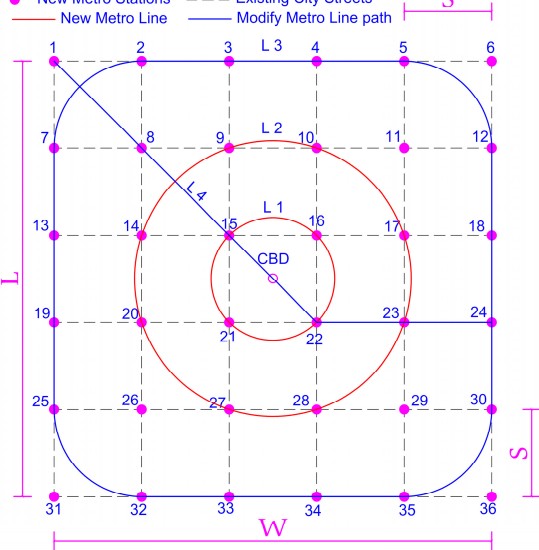

**Figure 9.** Ring-radial metro lines in the existing square city with one radial line (Case 6).

**Table 4.** Passenger Transfer OD Matrix of the ring-radial metro lines with modification of metro paths (Case 4).

| Station | 1 | 2 | 3 | 4 | 5 | 6 | 7 | 8 | 9 | 10 | 11 | 12 | 13 | 14 | 15 | 16 | 17 | 18 | 19 | 20 | 21 | 22 | 23 | 24 | 25 | 26 | 27 | 28 | 29 | 30 | 31 | 32 | 33 | 34 | 35 | 36 |
|---|---|---|---|---|---|---|---|---|---|---|---|---|---|---|---|---|---|---|---|---|---|---|---|---|---|---|---|---|---|---|---|---|---|---|---|---|
| 1 | 0 | 1 | 1 | 1 | 1 | 2 | 1 | 0 | 1 | 1 | 2 | 1 | 1 | 1 | 0 | 1 | 1 | 1 | 1 | 1 | 1 | 0 | 0 | 0 | 1 | ∞ | 1 | 1 | ∞ | 1 | ∞ | 1 | 1 | 1 | 1 | ∞ |
| 2 | 1 | 0 | 0 | 0 | 0 | 1 | 0 | 1 | 2 | 2 | 1 | 0 | 0 | 2 | 1 | 1 | 2 | 0 | 0 | 2 | 1 | 1 | 1 | 0 | 0 | ∞ | 1 | 2 | ∞ | 0 | ∞ | 0 | 0 | 0 | 0 | ∞ |
| 3 | 1 | 0 | 0 | 0 | 0 | 1 | 0 | 1 | 2 | 2 | 1 | 0 | 0 | 2 | 1 | 1 | 2 | 0 | 0 | 2 | 1 | 1 | 1 | 0 | 0 | ∞ | 1 | 2 | ∞ | 0 | ∞ | 0 | 0 | 0 | 0 | ∞ |
| 4 | 1 | 0 | 0 | 0 | 0 | 1 | 0 | 1 | 2 | 2 | 1 | 0 | 0 | 2 | 1 | 1 | 2 | 0 | 0 | 2 | 1 | 1 | 1 | 0 | 0 | ∞ | 1 | 2 | ∞ | 0 | ∞ | 0 | 0 | 0 | 0 | ∞ |
| 5 | 1 | 0 | 0 | 0 | 0 | 1 | 0 | 1 | 2 | 2 | 1 | 0 | 0 | 2 | 1 | 1 | 2 | 0 | 0 | 2 | 1 | 1 | 1 | 0 | 0 | ∞ | 1 | 2 | ∞ | 0 | ∞ | 0 | 0 | 0 | 0 | ∞ |
| 6 | 2 | 1 | 1 | 1 | 1 | 0 | 1 | 2 | 1 | 1 | 0 | 1 | 1 | 1 | 1 | 0 | 1 | 1 | 1 | 1 | 0 | 1 | 1 | 1 | 1 | ∞ | 0 | 1 | ∞ | 1 | ∞ | 1 | 0 | 1 | 1 | ∞ |
| 7 | 1 | 0 | 0 | 0 | 0 | 1 | 0 | 1 | 2 | 2 | 1 | 0 | 0 | 2 | 1 | 1 | 2 | 0 | 0 | 2 | 1 | 1 | 1 | 0 | 0 | ∞ | 1 | 2 | ∞ | 0 | ∞ | 0 | 0 | 0 | 0 | ∞ |
| 8 | 0 | 1 | 1 | 1 | 1 | 2 | 1 | 0 | 1 | 1 | 1 | 1 | 1 | 1 | 0 | 1 | 1 | 1 | 1 | 1 | 1 | 0 | 0 | 0 | 1 | ∞ | 1 | 1 | ∞ | 1 | ∞ | 1 | 1 | 1 | 1 | ∞ |
| 9 | 1 | 2 | 2 | 2 | 2 | 1 | 2 | 1 | 0 | 0 | 1 | 2 | 2 | 0 | 1 | 1 | 0 | 2 | 2 | 0 | 1 | 1 | 0 | 1 | 2 | ∞ | 0 | 0 | ∞ | 2 | ∞ | 1 | 2 | 2 | 2 | ∞ |
| 10 | 1 | 2 | 2 | 2 | 2 | 1 | 2 | 1 | 0 | 0 | 1 | 2 | 2 | 0 | 1 | 1 | 0 | 2 | 2 | 0 | 1 | 1 | 0 | 1 | 2 | ∞ | 0 | 0 | ∞ | 2 | ∞ | 1 | 2 | 2 | 2 | ∞ |
| 11 | 2 | 1 | 1 | 1 | 1 | 0 | 1 | 1 | 1 | 1 | 0 | 1 | 1 | 1 | 1 | 0 | 1 | 1 | 1 | 1 | 0 | 1 | 1 | 1 | 1 | ∞ | 0 | 1 | ∞ | 1 | ∞ | 1 | 0 | 1 | 1 | ∞ |
| 12 | 1 | 0 | 0 | 0 | 0 | 1 | 0 | 1 | 2 | 2 | 1 | 0 | 0 | 2 | 1 | 1 | 2 | 0 | 0 | 2 | 1 | 1 | 1 | 0 | 0 | ∞ | 1 | 2 | ∞ | 0 | ∞ | 0 | 0 | 0 | 0 | ∞ |
| 13 | 1 | 0 | 0 | 0 | 0 | 1 | 0 | 1 | 2 | 2 | 1 | 0 | 0 | 2 | 1 | 1 | 2 | 0 | 0 | 2 | 1 | 1 | 1 | 0 | 0 | ∞ | 1 | 2 | ∞ | 0 | ∞ | 0 | 0 | 0 | 0 | ∞ |
| 14 | 1 | 2 | 2 | 2 | 2 | 1 | 2 | 1 | 0 | 0 | 1 | 2 | 2 | 0 | 1 | 1 | 0 | 2 | 2 | 0 | 1 | 1 | 0 | 1 | 2 | ∞ | 0 | 0 | ∞ | 2 | ∞ | 1 | 2 | 2 | 2 | ∞ |
| 15 | 0 | 1 | 1 | 1 | 1 | 1 | 1 | 0 | 1 | 1 | 1 | 1 | 1 | 1 | 0 | 0 | 1 | 1 | 1 | 1 | 0 | 0 | 0 | 0 | 1 | ∞ | 1 | 1 | ∞ | 1 | ∞ | 1 | 1 | 1 | 1 | ∞ |
| 16 | 1 | 1 | 1 | 1 | 1 | 0 | 1 | 1 | 1 | 1 | 0 | 1 | 1 | 1 | 0 | 0 | 1 | 1 | 1 | 1 | 0 | 0 | 1 | 1 | 1 | ∞ | 0 | 1 | ∞ | 1 | ∞ | 1 | 0 | 1 | 1 | ∞ |
| 17 | 1 | 2 | 2 | 2 | 2 | 1 | 2 | 1 | 0 | 0 | 1 | 2 | 2 | 0 | 1 | 1 | 0 | 2 | 2 | 0 | 1 | 1 | 0 | 1 | 2 | ∞ | 0 | 0 | ∞ | 2 | ∞ | 1 | 2 | 2 | 2 | ∞ |
| 18 | 1 | 0 | 0 | 0 | 0 | 1 | 0 | 1 | 2 | 2 | 1 | 0 | 0 | 2 | 1 | 1 | 2 | 0 | 0 | 2 | 1 | 1 | 1 | 0 | 0 | ∞ | 1 | 2 | ∞ | 0 | ∞ | 0 | 0 | 0 | 0 | ∞ |
| 19 | 1 | 0 | 0 | 0 | 0 | 1 | 0 | 1 | 2 | 2 | 1 | 0 | 0 | 2 | 1 | 1 | 2 | 0 | 0 | 2 | 1 | 1 | 1 | 0 | 0 | ∞ | 1 | 2 | ∞ | 0 | ∞ | 0 | 0 | 0 | 0 | ∞ |
| 20 | 1 | 2 | 2 | 2 | 2 | 1 | 2 | 1 | 0 | 0 | 1 | 2 | 2 | 0 | 1 | 1 | 0 | 2 | 2 | 0 | 1 | 1 | 0 | 1 | 2 | ∞ | 0 | 0 | ∞ | 2 | ∞ | 1 | 2 | 2 | 2 | ∞ |
| 21 | 1 | 1 | 1 | 1 | 1 | 0 | 1 | 1 | 1 | 1 | 0 | 1 | 1 | 1 | 0 | 0 | 1 | 1 | 1 | 1 | 0 | 0 | 1 | 1 | 1 | ∞ | 0 | 1 | ∞ | 1 | ∞ | 1 | 0 | 1 | 1 | ∞ |
| 22 | 0 | 1 | 1 | 1 | 1 | 1 | 1 | 0 | 1 | 1 | 1 | 1 | 1 | 1 | 0 | 0 | 1 | 1 | 1 | 1 | 0 | 0 | 0 | 0 | 1 | ∞ | 1 | 1 | ∞ | 1 | ∞ | 1 | 1 | 1 | 1 | ∞ |
| 23 | 0 | 1 | 1 | 1 | 1 | 1 | 1 | 0 | 0 | 0 | 1 | 1 | 1 | 0 | 0 | 1 | 0 | 1 | 1 | 0 | 1 | 0 | 0 | 0 | 1 | ∞ | 0 | 0 | ∞ | 1 | ∞ | 1 | 1 | 1 | 1 | ∞ |
| 24 | 0 | 0 | 0 | 0 | 0 | 1 | 0 | 0 | 1 | 1 | 1 | 0 | 0 | 1 | 0 | 1 | 1 | 0 | 0 | 1 | 1 | 0 | 0 | 0 | 1 | ∞ | 1 | 1 | ∞ | 0 | ∞ | 0 | 0 | 0 | 0 | ∞ |
| 25 | 1 | 0 | 0 | 0 | 0 | 1 | 0 | 1 | 2 | 2 | 1 | 0 | 0 | 2 | 2 | 2 | 2 | 0 | 0 | 2 | 1 | 1 | 1 | 0 | 0 | ∞ | 1 | 2 | ∞ | 0 | ∞ | 0 | 0 | 0 | 0 | ∞ |
| 26 | ∞ | ∞ | ∞ | ∞ | ∞ | ∞ | ∞ | ∞ | ∞ | ∞ | ∞ | ∞ | ∞ | ∞ | ∞ | ∞ | ∞ | ∞ | ∞ | ∞ | ∞ | ∞ | ∞ | ∞ | ∞ | 0 | ∞ | ∞ | ∞ | ∞ | ∞ | ∞ | ∞ | ∞ | ∞ | ∞ |
| 27 | 1 | 1 | 1 | 1 | 1 | 0 | 1 | 1 | 0 | 0 | 0 | 1 | 1 | 0 | 0 | 0 | 0 | 1 | 1 | 0 | 1 | 0 | 1 | 1 | 0 | ∞ | 0 | 1 | ∞ | 1 | ∞ | 1 | 0 | 1 | 1 | ∞ |
| 28 | 1 | 2 | 2 | 2 | 2 | 1 | 2 | 1 | 0 | 0 | 1 | 2 | 2 | 0 | 1 | 1 | 0 | 2 | 2 | 0 | 1 | 1 | 0 | 1 | 2 | ∞ | 0 | 0 | ∞ | 2 | ∞ | 1 | 2 | 2 | 2 | ∞ |
| 29 | ∞ | ∞ | ∞ | ∞ | ∞ | ∞ | ∞ | ∞ | ∞ | ∞ | ∞ | ∞ | ∞ | ∞ | ∞ | ∞ | ∞ | ∞ | ∞ | ∞ | ∞ | ∞ | ∞ | ∞ | ∞ | ∞ | ∞ | ∞ | 0 | ∞ | ∞ | ∞ | ∞ | ∞ | ∞ | ∞ |
| 30 | 1 | 0 | 0 | 0 | 0 | 1 | 0 | 1 | 2 | 2 | 1 | 0 | 0 | 1 | 1 | 1 | 2 | 0 | 0 | 2 | 1 | 1 | 1 | 0 | 0 | ∞ | 1 | 2 | ∞ | 0 | ∞ | 0 | 0 | 0 | 0 | ∞ |
| 31 | ∞ | ∞ | ∞ | ∞ | ∞ | ∞ | ∞ | ∞ | ∞ | ∞ | ∞ | ∞ | ∞ | ∞ | ∞ | ∞ | ∞ | ∞ | ∞ | ∞ | ∞ | ∞ | ∞ | ∞ | ∞ | ∞ | ∞ | ∞ | ∞ | ∞ | 0 | ∞ | ∞ | ∞ | ∞ | ∞ |
| 32 | 1 | 0 | 0 | 0 | 0 | 1 | 0 | 1 | 2 | 2 | 1 | 0 | 0 | 2 | 2 | 2 | 2 | 0 | 0 | 2 | 1 | 1 | 1 | 0 | 0 | ∞ | 1 | 2 | ∞ | 0 | ∞ | 0 | 0 | 0 | 0 | ∞ |
| 33 | 1 | 0 | 0 | 0 | 0 | 0 | 0 | 1 | 1 | 1 | 0 | 0 | 0 | 1 | 1 | 0 | 1 | 0 | 0 | 1 | 0 | 1 | 1 | 0 | 0 | ∞ | 0 | 1 | ∞ | 0 | ∞ | 0 | 0 | 0 | 0 | ∞ |
| 34 | 1 | 0 | 0 | 0 | 0 | 1 | 0 | 1 | 2 | 2 | 1 | 0 | 0 | 2 | 2 | 2 | 2 | 0 | 0 | 2 | 1 | 1 | 1 | 0 | 0 | ∞ | 1 | 2 | ∞ | 0 | ∞ | 0 | 0 | 0 | 0 | ∞ |
| 35 | 1 | 0 | 0 | 0 | 0 | 1 | 0 | 1 | 2 | 2 | 1 | 0 | 0 | 2 | 2 | 2 | 2 | 0 | 0 | 2 | 1 | 1 | 1 | 0 | 0 | ∞ | 1 | 2 | ∞ | 0 | ∞ | 0 | 0 | 0 | 0 | ∞ |
| 36 | ∞ | ∞ | ∞ | ∞ | ∞ | ∞ | ∞ | ∞ | ∞ | ∞ | ∞ | ∞ | ∞ | ∞ | ∞ | ∞ | ∞ | ∞ | ∞ | ∞ | ∞ | ∞ | ∞ | ∞ | ∞ | ∞ | ∞ | ∞ | ∞ | ∞ | ∞ | ∞ | ∞ | ∞ | ∞ | 0 |

### 5.3. Capital Cost of Metro Lines

To design metro lines, it is necessary to understand the performance and cost characteristics of the different construction techniques of these lines. Metro lines were constructed with three techniques; underground line, aboveground line, and an elevated line. Moreover, the underground metro lines were constructed into two systems (single and twin). In this study, we develop an approximation equation to obtain the capital cost of twin underground lines, as presented in Equation (6).

The capital costs vary considerably with construction complexity and various construction techniques. The capital cost of metro systems is generally higher than that of other systems (bus, light rail transit, bus rapid transit, etc.). This is because of the need for rails, electrification, and signaling, stations, and higher vehicle costs.

For European projects, the value lies mainly between US $250–500 million (2020 prices). Alternatively, for the US projects, the range is US $250–750 million (2020 prices). The main reasons for the high variation in the route-kilometer costs are differences between plans as regards the ratio of the underground to aboveground construction, ground conditions, station spacing, type of rolling stock, environmental and safety constraints, and labor costs [71]. Moreover, the station type depends on the metro line system, so the construction method is considered the most crucial parameter on project costs. The station type includes the underground station, elevated station, and aboveground station. Generally, the total cost of the twin tunnels increases (30–80%) more than a single tunnel's total cost. The total cost of single or twin tunnels varies from one project to another according to tunnel diameter, the configuration of tunnels, overburden height, length of the tunnel between stations, construction quality, type of rolling stock, type of tunnel boring machine, etc. From previous projects constructed over all the world, and by using some software programs, an approximation equation is obtained to estimate an approximate cost of twin tunnels projects, respectively, as follows:

$$C_T = (60 - 80) \times Ln\left(2D_T^2\right) \times \left(0.55 + \frac{\zeta}{100}\right) \tag{6}$$

$$C_{Min} = 60 \times Ln\left(2D_T^2\right) \times 1.55 \tag{7}$$

$$C_{Max} = 80 \times Ln\left(2D_T^2\right) \times 1.55 \tag{8}$$

$$C_{avg} = 70 \times Ln\left(2D_T^2\right) \times 1.55 \tag{9}$$

where:

$C_T$   Capital cost for twin tunnel (2020-prices) (million per km-US$)
$C_{Min.}$ Minimum limit of capital cost (million per km-US$)
$C_{Max}$ Maximum limit of capital cost (million per km-US$)
$C_{avg}$ Average capital cost (million per km-US$)
$D_T$   External diameter of twin tunnel (m)
$\zeta$    Tunneling ratio (%)

According to previous equations, the average capital cost of twin tunnel metro, which is constructed underground with tunneling ratio ($\zeta$ = 0%, $\zeta$ = 100%), is presented in Figure 10.

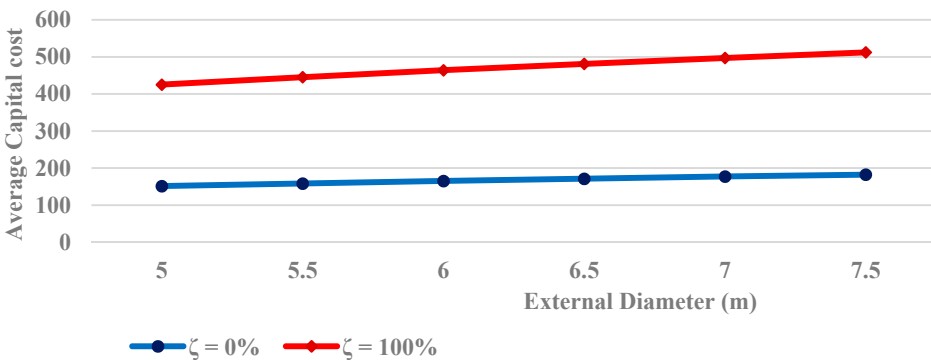

**Figure 10.** Capital average cost of twin tunnels (Million per km-2020 Prices-US$).

### *5.4. Accuracy of the Approximation Formula*

In order to establish the viability of the approximation formula, we first compare the average capital cost obtained by this formula with the actual capital cost that was previously presented by [71]. For consistency with previous work, the variation between average and actual cost was presented in Table 5.

**Table 5.** The capital cost of a twin tunnel project (Million per km-2020 Prices-US$).

| Project Name | Country | $\zeta$ (%) | $D_t$ (m) | Actual Costs | Avg. CapitalCosts | Variation. % |
|---|---|---|---|---|---|---|
| Copenhagen MetroPhases 1–3 | Denmark | 48% | 6.40 | 318 | 317.65 | −0.11 |
| Caracas Line 3 | Venezuela | 100 | 6.80 | 470 | 491.20 | +4.51 |
| Singapore metro | Singapore | 30 | 6.60 | 258 | 265.80 | +3.02 |
| Berlin U-Bahn metro | Germany | 100 | 6.60 | 459 | 484.70 | +5.60 |

## 6. Results and Discussion

All of the model cases cited above can improve the transit network in existing square cities by planning a new metro line in order to minimize passenger transfer numbers as well as the capital cost. An excellent way is to do this is with the control design of metro lines paths in two grid and ring-radial models.

### *6.1. Transfer Numbers Distribution of Transit Network*

As a result of the transit network connection, the distribution of passenger transfer numbers can be classified into zero-transfer ($TS_0$), one-transfer ($TS_1$), two-transfer ($TS_2$), and unserved ($TS_\infty$) (requiring more than two transfers) between OD pairs. The corresponding transfer numbers distribution of OD pairs satisfied with zero, one, or two transfers, as well as unserved, for each proposed case (Case1, Case2, . . . . . . , and Case 6) was presented in Figure 11.

Overall, all six cases seem to improve the overall transit connectivity of the existing square city based on the OD matrix cell distributions. However, the proposed objective function defines a weighted summation of coverage types ($\beta$ values) to control our preferences in the optimal case of the transit metro network. We recommend a set of $\beta$ structures in Table 6 to help choose the optimal case for each strategy.

These structures enable assessing the current state of the transit network. For instance, structure 1 concerns the not-connected stations in the transit network, whereas the second structure tries to minimize trips that need more than one transfer. The last structure is a balanced structure among the different types of transfer distribution.

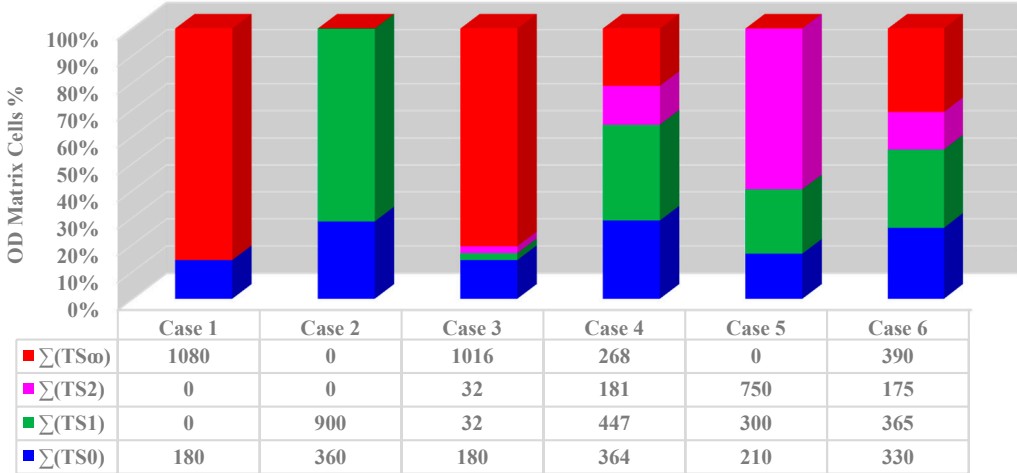

| | Case 1 | Case 2 | Case 3 | Case 4 | Case 5 | Case 6 |
|---|---|---|---|---|---|---|
| ∑(TS∞) | 1080 | 0 | 1016 | 268 | 0 | 390 |
| ∑(TS2) | 0 | 0 | 32 | 181 | 750 | 175 |
| ∑(TS1) | 0 | 900 | 32 | 447 | 300 | 365 |
| ∑(TS0) | 180 | 360 | 180 | 364 | 210 | 330 |

**Figure 11.** Each case effect the transfer distribution in the transit network.

**Table 6.** The weight of $\beta$ values for each selected structure.

| Structure No. | $\beta_0$ | $\beta_1$ | $\beta_2$ | $\beta_\infty$ |
|---|---|---|---|---|
| 1 | 0 | 0 | 0 | 1 |
| 2 | 0 | 0 | 1 | 1 |
| 3 | 0 | 1 | 2 | 9 |

In Figure 12, each proposed case effect on the PTN distribution values is evaluated according to the different strategies. For Structures 1 and 3, Cases 2 and 5 seem to dominate other cases with 100% served areas. Moreover, Case 2 achieves better performance regarding structure 2, with 100% served areas with one transfer.

Figure 13 shows the superiority of Case 4, with 23% direct trips between OD pairs. As direct trips are considered the main incentive for the passenger to use the public transit system, Case 4 is chosen to be the best metro transit network for the existing square city.

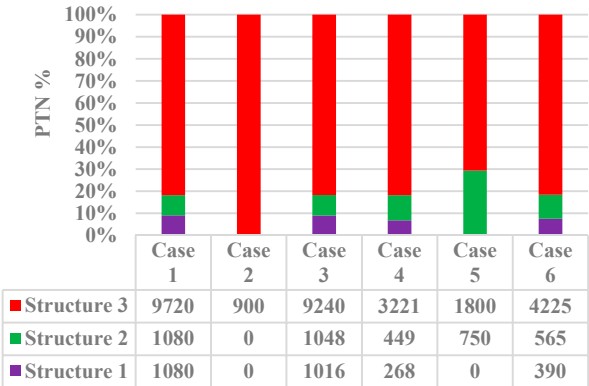

| | Case 1 | Case 2 | Case 3 | Case 4 | Case 5 | Case 6 |
|---|---|---|---|---|---|---|
| Structure 3 | 9720 | 900 | 9240 | 3221 | 1800 | 4225 |
| Structure 2 | 1080 | 0 | 1048 | 449 | 750 | 565 |
| Structure 1 | 1080 | 0 | 1016 | 268 | 0 | 390 |

**Figure 12.** PTN percent for six cases corresponding to each $\beta$ strategy.

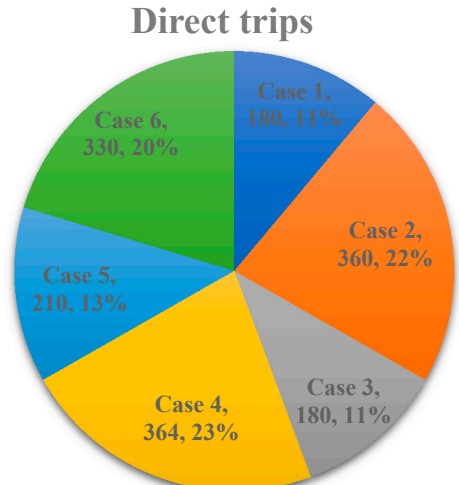

**Figure 13.** Direct trips percentage between stations in the transit network.

*6.2. Cost–Benefit Ratio (CBR)*

The CBR indicates the relationship between the cost and benefit of metro line construction or investment for analysis. It is shown by the present value of benefit expected divided by present value of cost, which helps determine the viability and value that can be derived from investment or project. The cost–benefit analysis reviews the overall value of a proposed project or initiative. Understanding the benefits of investing in a project is not always easily defined in revenues or monetary values. Some benefits are defined in qualitative terms, meaning how it impacts a specific community or group. When it comes to strategic business planning, a strategic plan often discusses the cost–benefit ratio in terms of a return on investments.

According to passenger transfer distribution, the results do not take into consideration the capital cost of metro lines for each case in order to select the optimal metro network (this is a limitation). Therefore, the capital cost must be taken into account so as to select the optimal network. In this subsection, we describe how to consider both passenger transfers as well as the capital cost to select the optimal metro network based on cost–benefit ratio (CBR).

A sequential selection process is then used to identify the optimal metro network by checking the maximum CBR. The order of optimal network selection is based on passenger transfer distribution with the capital cost of constructed metro lines. In each passenger transfer distribution, the procedure checks CBR for all cases of both models.

The CBR of the transit network is calculated by the passenger transfer distribution is multiplied by the capital cost of metro lines, as presented in Equation (10).

$$CBR = \frac{\sum PTN_{ij}}{\sum L \times C_{avg}} \tag{10}$$

In the formula,

$PTN_{ij}$ is the passenger transfer number distribution from station $i$ to station $j$

$L$ is metro length (km)

$C_{avg}$ is the average capital cost of twin tunnels ($C_{avg}$ = 0.485 billion per km-US$). The external diameter is $D_T$ = 6.60 m and tunnelling ratio is $\zeta$ = 100%.

In this subsection, we obtain the CBR for the transit network to select the optimal metro network based on direct trips as well as unserved OD pairs, as shown in Figure 14.

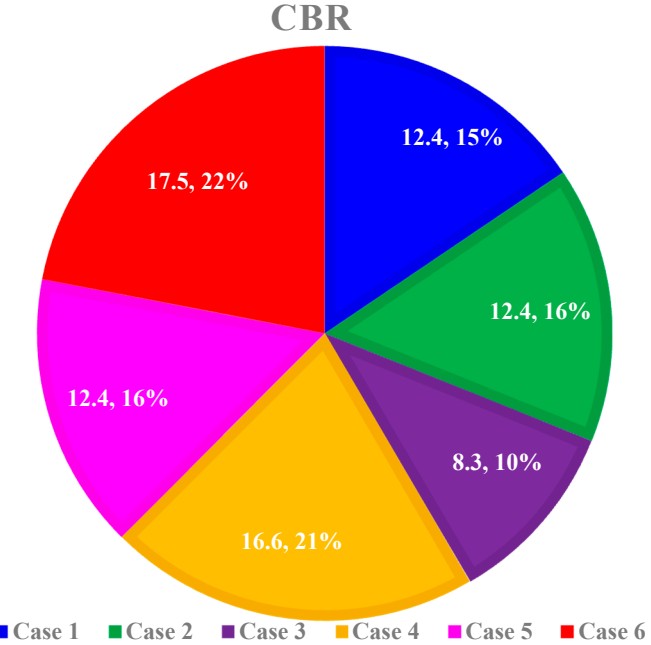

**Figure 14.** The CBR per billion US$ for six cases of both models in the transit network.

In Figure 14, Case 6 is the optimal metro network, which achieves the maximum CBR value (17.50 per billion US$) with a percentage of 22% in comparison to other cases.

We can see in Figure 13 that Case 4 is selected to be the optimal metro network in comparison with the results, which is shown in Figure 14. The optimal metro network is Case 6, which achieves passenger transfer distribution with a minimum capital cost.

*6.3. Discussion*

In this section, we describe the research findings related to the existing literature, and more details of the findings are presented in the conclusion section, considering the effect of the transfer numbers on a transit network design, particularly in existing cities. Furthermore, the transit network design was obtained in this research by proposing a metro network to increase the overall transit connectivity according to the following items:

1. Increasing direct trips between the stations.
2. Overall PTN reduction.
3. Benefit–cost ratio.

Compared with existing literature [9], Chen (2015) formulated two continuum approximation (CA) optimization models to design city-wide transit systems at minimum cost. Nevertheless, they assumed that the city streets form a grid for the grid metro network and ring-radial street for the ring-radial metro network, which is inadequate to design a ring-radial network in a grid street city. In this study, the ring-radial network's design in the existing square city with a grid street form was considered an advantage that was not discussed in previous studies.

To achieve comprehensive results for servicing existing cities of different sizes, this study idealized these cities with non-demand criteria. Additionally, the results indicated how to use these findings to achieve network planning of metro transit systems. In existing research, the authors studied the transit network design with consideration of travel time and passenger demand [72]; the transfer time composition on the total time expense of a transit trip [33]; the consideration of local route service and short-turn strategy [73]; the maximization of the net profit over a planning horizon [54]; feeder and non-feeder demand [74]; the bee colony optimization meta-heuristics [75,76]; the minimization of the total costs by incorporating various cost types [77]; the objectives of minimizing passengers' and

operators' costs [78–80]; travel time, monetary cost, number of transfers, and the total walking time [81]; and theimmune affinity model for emergency relief [82].

By comparing the above existing literature, the authors developed a genetic algorithm considering a non-demand criterion. The objective of the study for designing the optimal metro network is summarised as follows:

- Minimize passenger transfers between origin and destination, which leads to minimizing travel time.
- Minimize the total construction cost of metro lines by considering the cost–benefit ratio.

Moreover, this paper is valuable for designing metro transit systems in existing cities with no data available related to passenger demand and in new cities to forecast the optimal future metro line in the transit network.

The informal distribution of passenger trips lead to traffic congestion on the working days displayed the greatest spatial dependence, which tended to be regular and agglomerated. Congestion on weekends also displayed a certain degree of spatial dependence, with the evening peak exceeding the morning peak. The spatial dependence of traffic congestion on holidays was relatively weak, random, and scattered. Traffic congestion is responsible for many problems relating to long commuting times an declines in the quality of life [2,83–86]. Therefore, it is important to take into account the intensity of passenger flows, their direction, distribution by travel goals, and distribution over day time. It must maximize direct trips and minimize multiple passenger transfers between OD pairs.

The planning of the metro lines in existing cities is a complex problem. Therefore, it should take into account the availability of technological and territorial possibilities for placing metro lines in existing buildings according to the main three items: first, placing metro lines in main streets if possible; second, the safe distance between metro lines and the existing building; and third, the induced vibration of the tunnel boring machine (TBM) on an existing building during metro lines construction [87–91].

## 7. Conclusions

This study presents a new criterion for metro network design in existing square cities. The passenger transfer number (PTN) as a non-demand criterion is used to evaluate a set of proposed metro network cases of both models (grid and ring-radial network) to increase the transit network connectivity. The main objective of the non-demand criterion is to maximize the direct trips between origin and destination, which minimize the intensity of passenger flows of trips generated between each demand pair. However, the maximization of the demand coverage would still be the primary objective of any transit network design problem. An optimal metro network is chosen based on passenger transfer distribution with a minimum capital cost. The results show that the optimal metro network is a ring metro network with one radial line in which it increases the transit network connectivity and achieves the maximum CBR value (17.50 per billion US$) in comparison with other cases of both models. The optimal planning of the metro network has a direct relationship to the capital cost of metro lines. The ring-radial network was characterized by the lowest capital costs, in addition to the reduction in passengers' transfers in the transit network. The proposed methodology can be considered as a powerful tool that can help planners in generating quick and efficient solutions for designing the optimal metro network in the existing square city. This can be later assessed in more depth with operations research methods, in which in general the quality of their final solutions depends on the initial solution.

**Author Contributions:** M.O.: conceptualization, methodology, software, validation, and writing—original draft. A.S.A.: formal analysis, writing—editing. G.S.M.: methodology and reviewing. A.A.K.: Case study data collection, case study analysis, formulation, supervising, and reviewing. All authors have read and agreed to the published version of the manuscript.

**Funding:** This research received no external funding.

**Conflicts of Interest:** The authors declare no conflict of interest.

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
