# Peer review of "An Optimal Metro Design for Transit Networks in Existing Square Cities Based on Non-Demand Criterion"

_sustainability, doi:10.3390/su12229566_

Round 1

Reviewer 1 Report

The  presented topic of the study is certainly interesting from the point of view of transport planning, especially for cities where the question of forming a metro system arises. There have already been a lot of researches in this area and most of the approaches are based on the study of passenger supply and demand supply and demand for determining the location of metro lines. Thas why the presented non-demand approach is quite innovative and causes discussion. The article presents a clear methodology of the scientific approach based on mathematical modeling, which does not raise doubts about the reliability of the results. The authors ' literature review confirms a high level of knowledge and competence. However, it remains an open question to take into account the intensity of passenger flows, their direction, distribution by travel goals and distribution over day time. Also, the question arises how  to  take into account the availability of technological and territorial possibilities for placing metro lines in existing buildings, since the presented simplified matrixes do not provide a link to the territory, despite the fact that the metro is an underground transport type. In the Russian practice of transport planning, there is a theoretical approach to the formation of transport systems based on the analysis of correspondence matrices that are linked to the planning structure of the city: distances, time spent, types and nature of travel distribution are analyzed, which gives a reasonable result. However, the overall review of the article is positive. The article meets all requirements for scientific research, is performed at a high quality level, and presents the results of research in detail and reasonably. All discussion questions become prospective directions for continuing the development of the scientific topic. 

Author Response

Detailed responses to the reviewers' comments:

Our responses are presented in italic, bold font.

Reviewer 1:

" The presented topic of the study is certainly interesting from the point of view of transport planning, especially for cities where the question of forming a metro system arises. There have already been a lot of researches in this area and most of the approaches are based on the study of passenger supply and demand supply and demand for determining the location of metro lines. Thas why the presented non-demand approach is quite innovative and causes discussion. The article presents a clear methodology of the scientific approach based on mathematical modeling, which does not raise doubts about the reliability of the results. The authors ' literature review confirms a high level of knowledge and competence."

We really appreciate your opinion on our work.

However, it remains an open question to take into account the intensity of passenger flows, their direction, distribution by travel goals and distribution over day time.

Thank you for your valuable suggestion. In the revised manuscript, we clearly indicated that the selected non-demand criterion's main objective is to maximize the direct trips between origin and destination. This criterion of the design is based on both the topological layout of the transit network and the demand centers distribution and not depends on the uncertain information about the number of trips generated between each demand pair. We believe that would reflect on the intensity of passenger flows, their direction, distribution by travel goals, and distribution over daytime which, as mentioned, are uncertain information in nature.  However, the maximization of the demand coverage would still be the main objective of any transit network design problem( added in the conclusion section).

Moreover, we added a paragraph in the discussion section about the passenger flows distribution over day time.

Also, the question arises how to take into account the availability of technological and territorial possibilities for placing metro lines in existing buildings, since the presented simplified matrixes do not provide a link to the territory, despite the fact that the metro is an underground transport type. In the Russian practice of transport planning, there is a theoretical approach to the formation of transport systems based on the analysis of correspondence matrices that are linked to the planning structure of the city: distances, time spent, types and nature of travel distribution are analyzed, which gives a reasonable result.

Thank you for your point of argument. We agree with you that the planning of the metro lines in existing cities is a complex problem. So, it should be taken into account the availability of technological and territorial possibilities for placing metro lines in existing buildings. In the revised manuscript, we added a paragraph in the discussion section about the availability of technological and territorial possibilities for placing metro lines in existing buildings.

However, the overall review of the article is positive. The article meets all requirements for scientific research, is performed at a high quality level, and presents the results of research in detail and reasonably. All discussion questions become prospective directions for continuing the development of the scientific topic.

Thank you for your point of view. We again appreciate your opinion on our work.

Reviewer 2 Report

A very interesting paper defining network design for transit networks. This has been a forgotten topic in transport literature. I believe the paper is very important to the body of literature but requires some revision to clarify the following key concerns.

  • Definition of a "square city" is ambigious throughout the paper. There is mention that it a real "case study" is presented, however it seems to be a hypothetical grid city. This ambiguity leads to limitations in the value of the paper. If the model was tested on a real grid network (for example Washington DC) then there are many benefits but the issue is it seems to be applied to a generic network and ends up being more of a theoretical contribution. The theoretical contribution lacks strength because the optimisation programme is derived from an existing framework. 
  • Why is a "non-demand" criterion defined for the network design? This has not been clearly established in the comprehensive literature review presented but is very important to highlight the value of the formulation. If demand is ignored in the design, how would practitioners implement the design and manage potential future congestion issues? Perhaps I have misunderstood but it seems that the model is not capacity constrained which again limits its application.
  • There needs to be greater explanation of the mathematical formulation to clarify the objective function, constraints and the resulting impact. 
  • The results and discussion are good, but the cost-benefit analysis needs further explanation. The comparison with real cities is quite confusing at the moment because of the lack of clarity with the demonstration case study.

Overall, with a revision, the paper will be an important research study for transit network design. 

Author Response

Detailed responses to the reviewers' comments:

Our responses are presented in italic, bold font.

Reviewer 2:

" A very interesting paper defining network design for transit networks. This has been a forgotten topic in transport literature. I believe the paper is very important to the body of literature but requires some revision to clarify the following key concerns ".

We really appreciate your opinion on our work and hope that we have met your expectations.

  • Definition of a "square city" is ambigious throughout the paper. There is mention that it a real "case study" is presented, however it seems to be a hypothetical grid city. This ambiguity leads to limitations in the value of the paper. If the model was tested on a real grid network (for example Washington DC) then there are many benefits but the issue is it seems to be applied to a generic network and ends up being more of a theoretical contribution. The theoretical contribution lacks strength because the optimisation programme is derived from an existing framework.

Thank you for your point of view. We agree with you, and all have been modified to define a "square city" as a hypothetical grid city.

  • Why is a "non-demand" criterion defined for the network design? This has not been clearly established in the comprehensive literature review presented but is very important to highlight the value of the formulation. If demand is ignored in the design, how would practitioners implement the design and manage potential future congestion issues? Perhaps I have misunderstood but it seems that the model is not capacity constrained which again limits its application.

Thank you for your point. In the revised manuscript, we clearly indicated the objective of the "non-demand" criterion for transit network design ( We added a paragraph in the Literature review section). We emphasize that demand is not ignored; however, it is denoted by another robust criterion. In metro cases, capacity could be treated easily (unlike bus networks) in the operational stage by setting the train vehicle number.

We should also emphasize that our research method is new, how to develop a new metro network with a high level of connectivity, and at the same time, escape from the combinatorial complexity of line design problem and the uncertainty of demand evaluation. We believe that our approach is capable of doing that for real-large case studies. We also believe that the proposed methodology is practical, new, and easy to follow and have a significant contribution to the field.

  • There needs to be greater explanation of the mathematical formulation to clarify the objective function, constraints and the resulting impact.

Thank you for your valuable suggestion. In the revised manuscript, a separate paragraph at the end of section 3 has been added to highlight the explanation of the mathematical formulation. Where this study presents an analytical model of metro network design using a non-demand criterion objective function of the Passenger Transfer Number (PTN) for improving the coverage performance of the existing city networks. The explanation of the mathematical formulation could be summarized in the following points;

  • Introducing a mathematical formulation for determining the PTN of the metro transit network.
  • Obviating the combinatorial complexity of transit routing with an efficient, straightforward scheme of design.
  • Presenting the mathematical notations in a general framework which gives the possibility of generalizing the methodology in other studies or even using it as a subroutine in presenting more sophisticated methodologies.
  • Developing an exclusive non-demand criterion of metro network design., unlike reviewed studies where they all are demand-oriented.

  • The results and discussion are good, but the cost-benefit analysis needs further explanation. The comparison with real cities is quite confusing at the moment because of the lack of clarity with the demonstration case study.

We again appreciate your help and assistance. We explained the cost-benefit analysis by adding a new paragraph in the Cost-Benefit Ratio subsection.

Overall, with a revision, the paper will be an important research study for transit network design.

Thank you for your point of view. We again appreciate your opinion on our work.

Reviewer 3 Report

The work shows the theoretical approach to transport network planning.
The model used is very simplified, but it allows drawing preliminary conclusions for planners.
I am afraid that in real conditions the planners would not allow such large simplifications.
Another problem with the ring metro is the lack of starting stops where compensatory stops would take place, hence such systems are particularly vulnerable to delays and the lack of cyclicality of train arrivals.

The descriptions of the formulas (at the beginning in chapter 3) should be corrected as they are completely irredible. Recommend:

where:
TS0 - ......
alphals - ....
etc,

and finally (preferably also in the form of a graph) the method of selecting the coefficients.

Lines ~400 where the algorithm is described. It is worth presenting it in the form of a graph, not purely written.

Table 1 should be placed after the esplanation in the text as the reader is not able to understand what it is about.

Author Response

Detailed responses to the reviewers' comments:

Our responses are presented in italic, bold font.

Reviewer 3:

" The work shows the theoretical approach to transport network planning.

The model used is very simplified, but it allows drawing preliminary conclusions for planners. I am afraid that in real conditions the planners would not allow such large simplifications. Another problem with the ring metro is the lack of starting stops where compensatory stops would take place, hence such systems are particularly vulnerable to delays and the lack of cyclicality of train arrivals.:"

We hope that the revised manuscript meets your recommendation and suggestions. A major revision has been made according to the raised points. Also, we should emphasize that the manuscript presents a significant practical contribution (that is always missing in this type of research problem) along with some theoretical development in the selected objective function and software implementations. Moreover, we added new paragraphs in the Literature review section with the application of the non-demand criterion in real conditions of Greater Cairo city in Egypt country.

We should also emphasize that our research method is new, how to develop a new metro network with a high level of connectivity, and at the same time, escape from the combinatorial complexity of line design problem and the uncertainty of demand evaluation. We believe that our approach is capable of doing that for real-large case studies. We also believe that the proposed methodology is practical, new, and easy to follow and have a significant contribution to the field.

The descriptions of the formulas (at the beginning in chapter 3) should be corrected as they are completely irredible. Recommend:

where:

TS0 - ......

alphals - ....

etc,

and finally (preferably also in the form of a graph) the method of selecting the coefficients..

Thank you for your remark. In the revised manuscript, The description of the formulas has been modified according to your valuable suggestions.

Lines ~400 where the algorithm is described. It is worth presenting it in the form of a graph, not purely written.

Thank you for the suggestion. In the revised manuscript, the algorithm is described in the form of a graph according to your valuable suggestions

Table 1 should be placed after the explanation in the text as the reader is not able to understand what it is about.

Thank you for the suggestion. In the revised manuscript, Table 1 was placed after the explanation in the text according to your valuable suggestions

Round 2

Reviewer 2 Report

All my concerns have been addressed - this is now a fantastic paper and happy for it to be published.